# Altered projection-specific synaptic remodeling and its modification by oxytocin in an idiopathic autism marmoset model
Jun Noguchi [1] ✉, Satoshi Watanabe [1], Tomofumi Oga[1], Risa Isoda[1], Keiko Nakagaki[1], Kazuhisa Sakai[1], Kayo Sumida[2], Kohei Hoshino[3], Koichi Saito[2], Izuru Miyawaki[3], Eriko Sugano [4], Hiroshi Tomita [4], Hiroaki Mizukami[5], Akiya Watakabe[6], Tetsuo Yamamori[6,7,8] & Noritaka Ichinohe [1] ✉

Alterations in the experience-dependent and autonomous elaboration of neural circuits are assumed to underlie autism spectrum disorder (ASD), though it is unclear what synaptic traits are responsible. Here, utilizing a valproic acid–induced ASD marmoset model, which shares common molecular features with idiopathic ASD, we investigate changes in the structural dynamics of tuft dendrites of upper-layer pyramidal neurons and adjacent axons in the dorsomedial prefrontal cortex through two-photon microscopy. In model marmosets, dendritic spine turnover is upregulated, and spines are generated in clusters and survived more often than in control marmosets. Presynaptic boutons in local axons, but not in commissural long-range axons, demonstrate hyperdynamic turnover in model marmosets, suggesting alterations in projection-specific plasticity. Intriguingly, nasal oxytocin administration attenuates clustered spine emergence in model marmosets. Enhanced clustered spine generation, possibly unique to certain presynaptic partners, may be associated with ASD and be a potential therapeutic target.

Autism spectrum disorder (ASD) is a developmental disorder characterized by impairments in social communication, social interaction, and stereotyped behaviors[1,2]. Individuals with ASD often have learning disabilities and have difficulty learning to recognize verbal or non-verbal social information[3]. Proper refinement of neural networks during learning is achieved by coordinated synaptic remodeling, which may be altered in ASD. Dendritic spines, which are postsynaptic protrusions that receive most excitatory inputs[4–6], have been observed longitudinally using two-photon microscopy in mouse models of ASD. This has made it possible to explore the spatiotemporal characteristics of synaptic remodeling in ASD. Morphological analysis of dendrites with in vivo two-photon microscopy has shown that accelerated spine generation and elimination in the motor and primary sensory cortices is a consistent phenotype in numerous ASD mouse models (BTBR, 15q11–13 duplication, Neuroligin mutant, *FMR1* knockout, *MeCP2* duplication, etc.)[7–10].

Clustered generation of postsynaptic dendritic spines, in which new spines form in close proximity to one another, plays a critical role in learning and memory[11–17]. Training increases clustered spine generation, which has been found to be correlated with learning performance in corresponding brain regions[10,18]. Neuronal modeling studies have suggested that enhanced clustered spine generation increases memory discrimination and the storage capacity of neuronal networks[17,18]. Excessively clustered spines may contribute to the development of ASD symptoms, but little is known about their involvement in this condition. On the other hand, it has been reported that diversity of excitatory synapses includes several synapse types or subtypes defined by molecular and other characteristics, and that certain circuits or connectome networks prefer particular types of synapses[19]. Changes in gene expression in ASD may alter the plasticity of specific cortical projections, which may in turn perturb the formation of neural circuits adapted to learning.

[1]Department of Ultrastructural Research, National Institute of Neuroscience, National Center of Neurology and Psychiatry, Kodaira, Japan. [2]Environmental Health Science Laboratory, Sumitomo Chemical Co., Ltd., Osaka, Japan. [3]Preclinical Research Laboratories, Sumitomo Pharma Co., Ltd., Osaka, Japan. [4]Laboratory of Visual Neuroscience, Graduate Course in Biological Sciences, Iwate University, Morioka, Japan. [5]Division of Genetic Therapeutics, Jichi Medical University, Shimotsuke, Japan. [6]Laboratory for Molecular Analysis of Higher Brain Function, Center for Brain Science, RIKEN, Wako, Japan. [7]Present address: Laboratory for Haptic Perception and Cognitive Physiology, Center for Brain Science, RIKEN, Wako, Japan. [8]Present address: Department of Marmoset Biology and Medicine, CIEM, Kawasaki, Japan. ✉e-mail: jnoguchi@ncnp.go.jp; nichino@ncnp.go.jp

However, projection-specific variations of synapse remodeling in ASD have not been explored.

The common marmoset (*Callithrix jacchus*), a small New World monkey, has attracted considerable attention due to its rich repertoire of social behaviors, a well-developed prefrontal cortex (PFC) that supports high-level social ability, and gene expression networks that are similar to those in humans[20]. In fact, marmosets are more similar to humans than rodents in terms of their synaptic proteome[21]. We previously developed a valproic acid (VPA)–induced ASD model in the common marmoset[22]. VPA is an antiepileptic drug and also functions as a histone deacetylase inhibitor that may epigenetically increase the risk of ASD in humans by suppressing histone deacetylases in the fetal brain. Despite the fact that VPA administration was the sole environmental factor contributing to the development of ASD in this model, and there were no genetic variations, it was sufficient to induce gene expression changes in juvenile marmosets that were more typical of human idiopathic ASD than those in monogenetic ASD rodent models[22]. The characteristics of VPA-exposed marmosets suggest that these ASD models may be useful for bridging rodent ASD studies and human ASD.

A notable portion of individuals with ASD persists in experiencing a range of challenges such as anxiety or depression in their daily lives into adulthood[23,24]. It is critical to understand ASD pathophysiology in adult ASD model animals and explore treatments. In this study, we investigated the temporal remodeling of neural circuits using in vivo two-photon longitudinal imaging in VPA-exposed adult marmosets. We analyzed synaptic dynamics at 3-day intervals in the apical tuft dendrites of pyramidal neurons in the primate-specific dorsomedial PFC (dmPFC). The dmPFC is involved in social cognition and habit formation, and was found to exhibit less activity in nonverbal information–biased judgment in ASD individuals[25]. We also observed the axonal boutons of local and long-distance cortical callosal connections labeled with fluorescent proteins of different colors to investigate whether circuit-specific synaptic remodeling was altered in the ASD model animal. Our study revealed that in VPA-exposed marmosets, turnover of postsynaptic dendritic spines was upregulated, and spines were actively generated in clusters but seemed to be randomly eliminated. Notably, in the marmoset model, the rates of both generated clustered spines and subsequently surviving spines (carryover spines) were much higher than in controls (3.3 and 5.9 fold, respectively). Presynaptic boutons of local axons, but not long-range commissural axons, showed more frequent turnover in model marmosets than in controls. Importantly, the clustering propensity of emergent spines in VPA-exposed marmosets was reduced by intranasal administration of oxytocin, which is currently being tested as a therapy for human ASD. These results suggest that both enhanced generation of clustered spines and altered turnover of local neuronal circuits are associated with ASD. We consider that these processes may be potential therapeutic targets for ASD symptoms.

## Results

### Two-photon imaging of the dendrites of PFC layer-2/3 pyramidal neurons in living marmosets

Changes in spine geometry (density) have been observed in the brains of individuals with ASD[26], and alterations in synaptic electrophysiological characteristics have been reported in animal models of ASD[22,27]. In the present study, we used a marmoset ASD model to investigate spine dynamics in the dmPFC. We used three adult VPA-exposed marmosets and four adult unexposed (UE) marmosets (See Methods and Fig. 1a). We employ the prenatal VPA exposure model due to its relevance to in utero VPA exposure in humans and a rich accumulation of findings across multiple experimental animal species. We inoculated the marmoset dmPFC with adeno-associated virus (AAV) and expressed fluorescent proteins mainly in layer-2/3 pyramidal neurons[28]. These proteins showed sufficient fluorescence for in vivo two-photon imaging. The fluorescent protein–expressing neurons were locally distributed with an average diameter of 1.81 mm (range: 1.02–2.87) axially and 1.53 mm (range: 0.86–2.21) laterally. Post-experiment immunohistochemistry using antibodies against

Iba-1 and GFAP showed no obvious increases in activated microglia or astrocytes, respectively (Supplementary Fig. 1).

We performed time-lapse observation of the dmPFC using a two-photon microscope[28,29] (Fig. 1a, b). Every 3 days, we conducted three imaging sessions of identical apical tuft dendrites in UE and VPA-exposed animals (*n* = 14 and 12 dendrites, respectively; mean dendrite span per branchlet: 31.7 ± 5.9 μm and 31.6 ± 5.8 μm, respectively). We found no significant differences in the dendritic spine density on apical tuft dendrites between the two types of marmosets (Fig. 1c). We next compared the spine generation and elimination rates between the VPA-exposed and UE groups (Fig. 1b, d). We found that the rates of both spine generation and elimination were approximately two times higher in VPA-exposed than UE marmosets (Fig. 1d). This result was consistent with the results from various ASD mouse models of enhanced spine generation and elimination[7–10].

### Smaller spines in the ASD model marmoset were more prone to elimination

We next performed an analysis of spine volume, which is a crucial measure of synaptic weight. We calculated the normalized spine volume by dividing each spine's fluorescent intensity with the dendrite shaft intensity, and pooled the results of all dendrites. The results showed no significant difference in the distribution of spine volume between the VPA-exposed and UE groups (Fig. 1e). As in previous studies[30,31], both groups showed a significant difference in volume distribution between spines that were eliminated during the 3-day period and those that survived (Fig. 1f). The volume distribution of the VPA-exposed and UE groups did not differ when the eliminated and surviving groups were analyzed separately (Fig. 1f); however, for the smaller spines, the proportion of eliminated spines was significantly larger in the VPA-exposed group (Fig. 1g). These results suggest that the synaptic weight distributions in the dendrites were similar in the two types of marmosets, whereas the smaller spines were more vulnerable in VPA-exposed animals.

### Newly generated spines clustered more frequently in VPA-exposed marmosets than in controls

The generation of spines in close proximity to each other, or clustered spine generation, is considered to have functional significance, especially in learning and memory[5,10,15,18]. We next examined whether the generated spines in the VPA-exposed and UE marmosets were clustered or not (Fig. 2a). We chose a 3 μm window for our analyses, since several biochemical, physiological, and structural studies have suggested that a 3–10 μm distance between spines facilitates sharing of resources, spine co-activation, and learning-induced structural plasticity[32–38]. Moreover, the difference between the UE and VPA-exposed animals in spine clustering reached a plateau at a distance of 3 μm between spines, suggesting that events occurring within 3 μm are particularly important (Supplementary Fig. 2a, Methods). By using this threshold value, we found that clustered spine generation occurred 3.3 times more frequently in VPA-exposed animals than in UE animals (Fig. 2b), even though the total number of generated spines in VPA-exposed marmosets was only about twice as many as that in the UE marmosets (Fig. 1d). The clustered generated spines even accounted for 6.2% of the total spines on the dendrites. Next, we created a cumulative plot of the distance between generated spines. As shown in Fig. 2c, there was a significant difference in the distribution of interspine distances between the VPA-exposed and UE groups. Again, the difference between the VPA-exposed and UE animals in terms of the probability of clustered spine generation was large (2.6 fold; Fig. 2c). These results suggest the existence of a mechanism by which dendritic spines actively appear in clusters in the animals exposed to VPA. Therefore, we conducted a Monte Carlo simulation experiment to determine whether there was more clustering bias in VPA-exposed marmosets than in the hypothetical uniform random spine distribution (Fig. 2d). To prevent underestimation of the inter-spine distances between the newly generated spines, we conducted a simulation with all measured dendrites concatenated into one long dendrite, where dendrite lengths and spine counts of each dendrite were summed. We

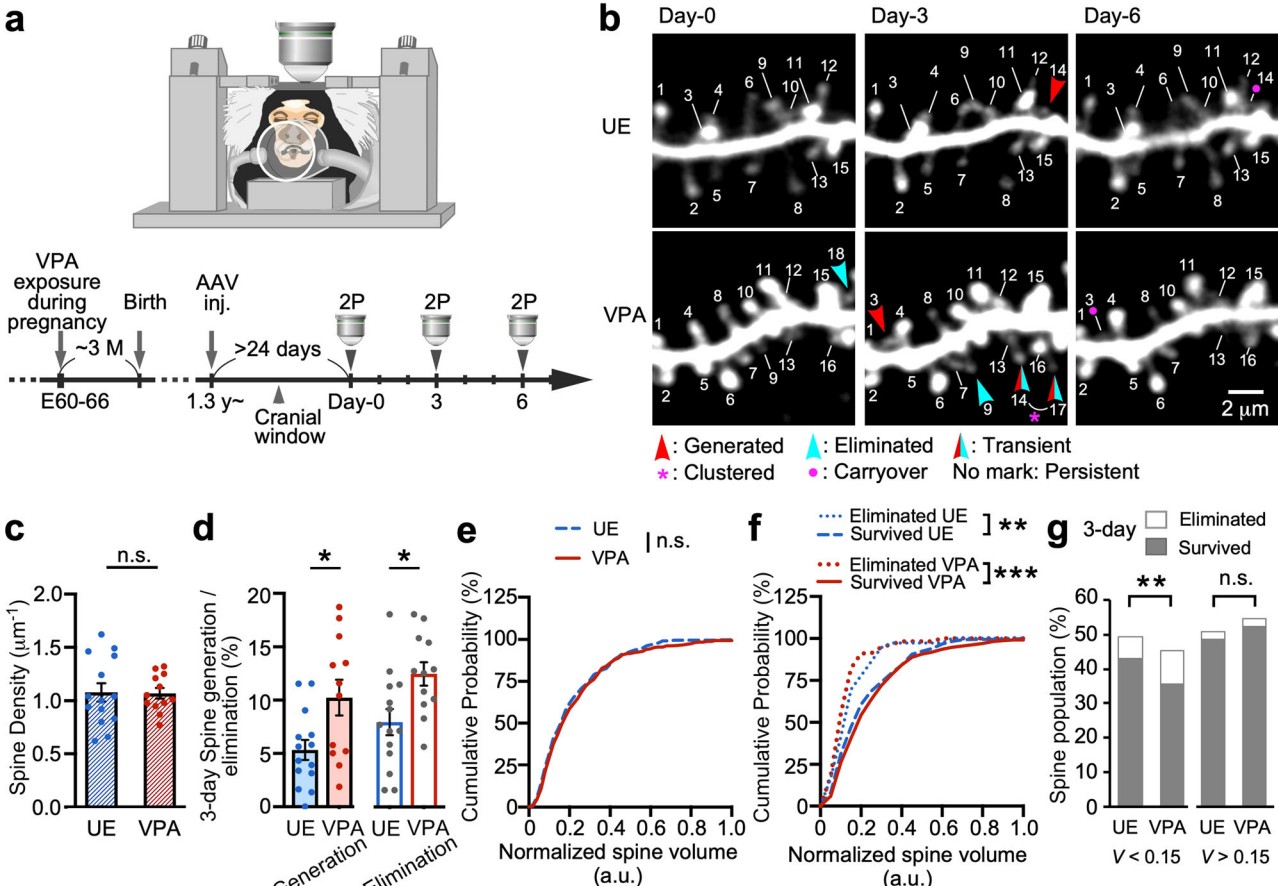

**Fig. 1 | In vivo two-photon imaging of mature marmoset dendrites. a** A marmoset was anesthetized for two-photon (2P) imaging. The lower panel shows the experimental schema. **b** Longitudinal 2P imaging from the PFC layer-2/3 pyramidal neuron tuft dendrites. Every 3 days, the same dendrites were imaged in UE (unexposed; control animals) and VPA-exposed (ASD model) marmosets, and the same spines were labeled using the same numbers throughout the period. **c** Mean spine densities of dendrites (mean ± s.e.m.; $P = 0.85$, Mann–Whitney U test; $n = 14$ and 12 dendrites in four and three UE and VPA-exposed animals, respectively). **d** Three-day spine generation rates (mean ± s.e.m.; $P = 0.031$, Mann–Whitney U test; the same dendrites as in **c**), and 3-day elimination ($P = 0.012$). **e** Cumulative plots of the normalized spine volume distribution ($P = 0.49$; Kolmogorov–Smirnov test; $n = 480$ and 473 spines in UE and VPA-exposed animals, respectively). **f** Cumulative plots of eliminated and surviving spines in UE animals ($P = 0.0035$; $n = 36$ and 444 spines, respectively), and in VPA-exposed animals during the 3-day observation ($P < 0.001$; $n = 56$ and 417 spines, respectively). **g** The population of eliminated and surviving spines during the 3-day observation are shown separately for groups with a spine volume of $V < 0.15$ and $V > 0.15$ ($V < 0.15$; $P = 0.0019$; Fisher's exact test; $n = 25, 46, 210,$ and 169 spines for eliminated UE, eliminated VPA-exposed, surviving UE, and surviving VPA-exposed spines, respectively) ($V > 0.15$; $P = 0.83$; $n = 11, 10, 234,$ and 248 spines, respectively). ***$P < 0.001$; **$P < 0.01$; *$P < 0.05$; n.s. not significant.

explored the impact of adjusting cluster thresholds to 3, 6, and 9 microns on the outcomes of spine generation and elimination clustering (Supplementary Fig. 2c–p). The clustering probability of the generated spines actually observed in VPA-exposed animals corresponded to a probability of $P = 0.00042$ for the distribution of clustering probabilities predicted by the simulation at the 3 μm threshold (Supplementary Fig. 2d). This P-value is much smaller than the probabilities at the 6 μm ($P = 0.016$) or 9 μm ($P = 0.29$) thresholds (Supplementary Fig. 2f, h). To investigate this further, we plotted the actual clustering probabilities against the simulated clustering probability, and again found that the difference between chance-level clustering and observed clustering was greatest at the 3 μm threshold that provides the Youden's index in the chart (Supplementary Fig. 2o)[39]. Since the specific clustering bias disappears at the 9 μm threshold, indicating a physiological significance of the interaction within this distance. This analysis elucidates that the threshold set at 3 μm offers a good cut-off value for assessing spine clustered generation. In contrast, the clustering probability of the generated spine pairs was remained below the 95th percentile in the UE group (Fig. 2e). On the other hand, the specific clustering of eliminated spines was not as prominent as generated spines either in UE and VPA-exposed animals (Supplementary Fig. 2i–n, p, and 3). Although there is a disparity in the effect of distance on clustered spine elimination between UE

and VPA-exposed animals (Supplementary Fig. 3c), this effect was considerably less pronounced than spine generation (Supplementary Fig. 2b). The slight increase in clustered elimination observed in VPA-exposed marmosets could stem from heightened nonspecific clustering associated with the larger number of eliminated spines. Simulation analysis revealed that the clustering bias of spine elimination in VPA-exposed animals is not significant (Supplementary Fig. 2p, 3d, e).

Furthermore, an analysis of the crosstalk between spine generation and elimination was conducted, as depicted in Supplementary Fig. 4. Cumulative frequency distributions were generated for the distances between newly generated spines and their nearest eliminating spines. Consequently, no statistically significant difference was observed between the UE and VPA-exposed marmosets in this regard (Supplementary Fig. 4). Considering that the generated clustered spines may appear on specific dendrites, we conducted another simulation in which the dendrite length and the number of spines generated on each dendrite were left unaltered (that is, each dendrite was treated separately without concatenation), and obtained similar results (Supplementary Fig. 5). We conclude that the clustering of newly generated spines, but not that of eliminated spines, on the dendrites of VPA-exposed marmosets is substantially enhanced rather than randomly distributed.

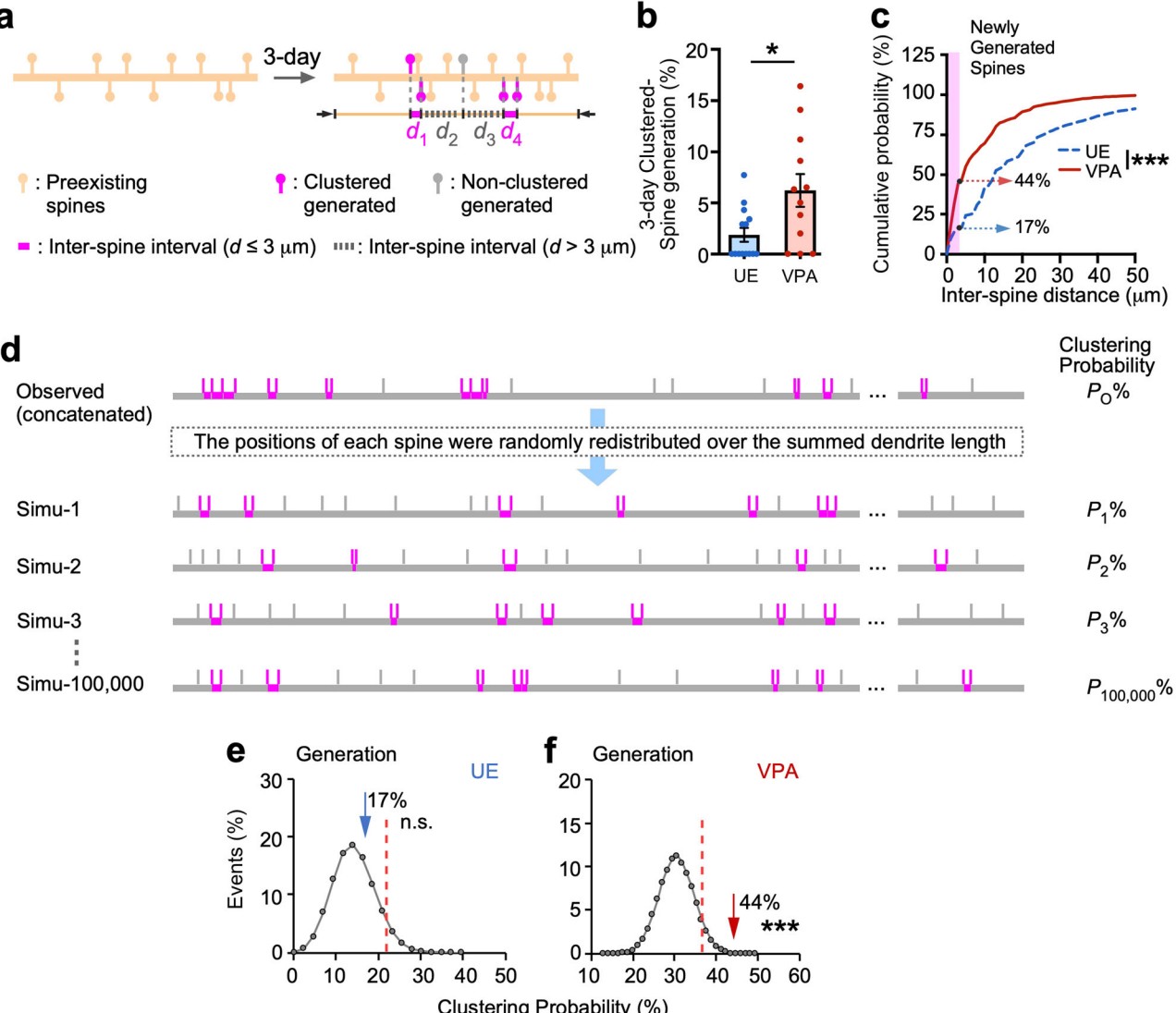

**Fig. 2 | Dendritic spine clustered generation is more predominant in VPA-exposed marmosets. a** Schematic drawing of clustered spine generation. Pairs of newly generated spines were considered to be clustered if they occurred within 3 μm of each other. Spines colored magenta or gray represent clustered and non-clustered spines, respectively. **b** Comparison of the percentage of clustered generated spines to the total number of spines between UE and VPA-exposed animals (mean ± s.e.m.; $P = 0.026$, Mann–Whitney U test; $n = 14$ and 12 dendrites in four UE and three VPA-exposed animals, respectively). **c** Distribution of inter-spine distances between newly generated spines ($P = 0.00048$; Kolmogorov–Smirnov test; $n = 43$ and 85 spine pairs in UE and VPA-exposed animals, respectively). Magenta-shaded area indicates inter-spine distances shorter than 3 μm, and numbers indicate the probabilities of clustering (within 3 μm). To prevent underestimation of the inter-spine distance, dendrites were concatenated into one long dendrite. **d** Validation of clustering bias by Monte Carlo simulation. In the simulation, the new spine positions were randomly determined with a uniform distribution over the summed dendrite length. Clustering probabilities for all inter-spine distances were calculated for each simulation, and the distributions of the clustering probability from 100,000 iterations are shown in **e** and **f**. **e, f** Circles connected with gray lines represent probability plots of clustering events from 100,000 simulations; the actual numbers of spine clusters are represented by arrows ($P = 0.24$ and 0.00042; $n = 44$ and 86 newly generated spines in 14 and 12 dendrites in UE and VPA-exposed animals, respectively). Dotted red lines show 95th percentiles. ***$P < 0.001$; *$P < 0.05$; n.s. not significant.

## The carryover fraction of newly generated clustered spines was higher in VPA-exposed marmosets than in controls

Three consecutive imaging sessions allowed us to monitor the fate of preexisting and newly generated spines over 3 days[9]. The carryover spines were defined as the newly generated spines that survived to the last session. In other words, carryover spines are spines whose presence was denied by the first observation and confirmed by the second and third observations (Fig. 3a, b; GS). As discussed below, the survival rate of carryover spines is not different between UE and VPA-exposed animals. However, since the fraction of newly generated spines are two times higher in VPA-exposed animals (Fig. 1d), the fraction of carryover spines was two times higher in VPA-exposed marmosets than in UE marmosets (Fig. 3c). We further divided the newly generated spines into clustered and non-clustered spines and computed their carryover fractions. The carryover fraction of clustered-generated spines in VPA-exposed marmosets was 5.9 times higher than that in UE marmosets (Fig. 3d). On the other hand, there was no difference in the carryover fraction of non-clustered spines with or without VPA treatment (Fig. 3d). Thus, clustered spines accounted for a greater proportion of spine carryover in the model marmosets than in the UE marmosets.

By calculating the carryover fraction, we can understand how newly generated spines contribute to the overall synaptic population. Next, by calculating the spine survival rate, we can focus on the persistence of each newly generated and pre-existing spine, offering a more targeted measure of synaptic resilience. Spine survival rate was defined as the percentage of spines present at the second observation that were still present at the third observation (Fig. 3b). Both in the VPA-exposed and UE

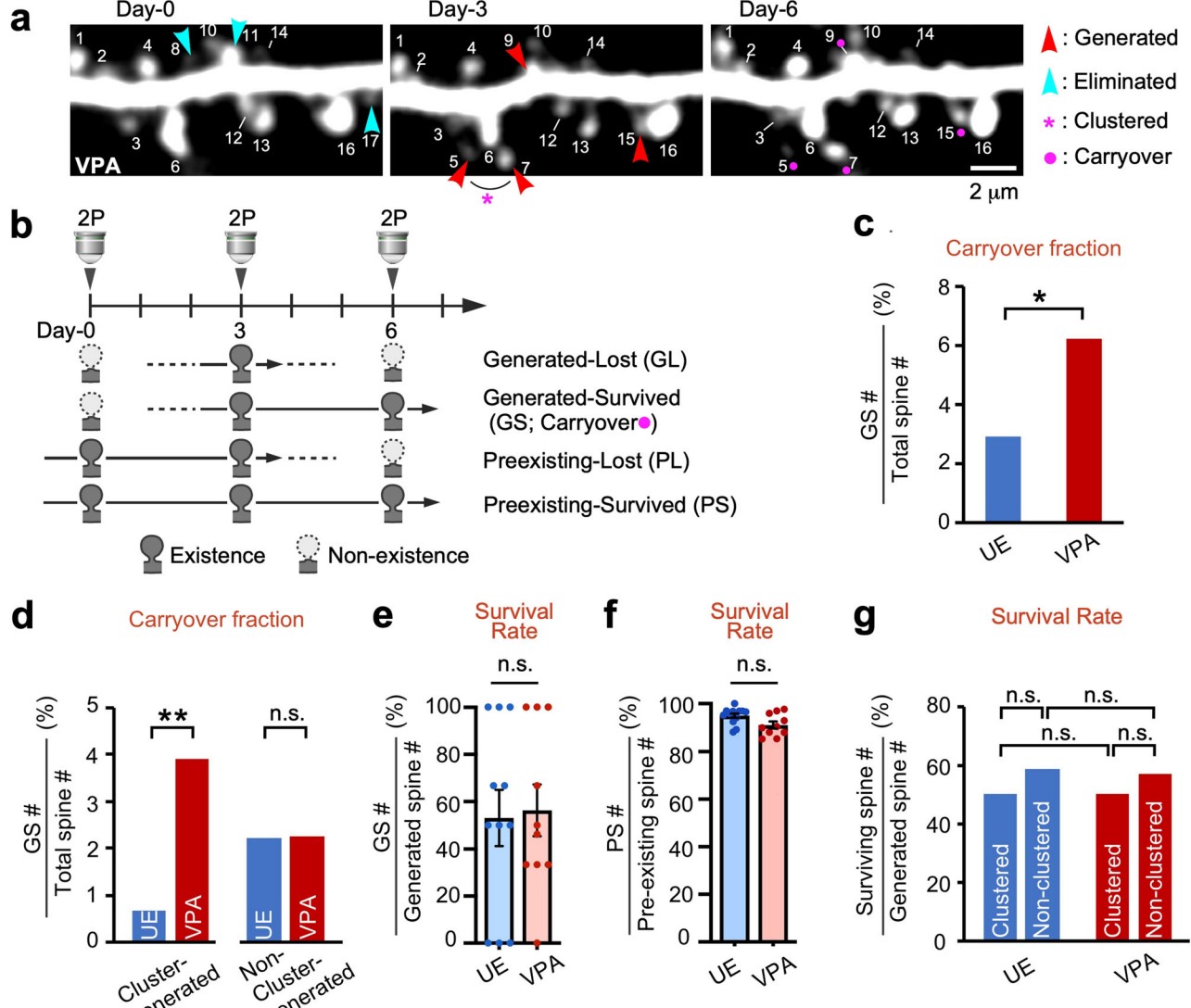

**Fig. 3 | The surviving fraction of newly generated spines (carryover spines) is much larger in VPA-exposed marmosets than in controls, although spine stability is similar. a** Representative dendrite images taken every 3 days are presented as in Fig. 1b. **b** A diagram showing four patterns of spine generation and elimination. The red dot in the legend indicates the carryover spines labeled with a red dot in Figs. 1b and 3a. **c** The ratio of the surviving fraction of newly formed spines (carryover spines) to the total spines was larger in VPA-exposed animals than in UE animals. ($P = 0.035$, Fisher's exact test; $n = 13$, 448, 22, and 355 for UE carryover spines, UE total spines, VPA-exposed carryover spines, and VPA-exposed total spines, respectively). **d** Among clustered generated spines, the ratio of the carryover spine fraction to the total spines was much larger in VPA-exposed animals than in UE animals ($P = 0.0020$, Fisher's exact test; $n = 3$, 448, 14, and 355 for UE carryover, UE total, VPA-exposed carryover, and VPA-exposed total spines, respectively). By contrast, the carryover spine fraction was not significantly different among non-

clustered spines ($P > 0.99$, Fisher's exact test; $n = 10$, 448, 8, and 355 for UE carryover, UE total, VPA-exposed carryover, and VPA-exposed total spines, respectively). **e** The survival rate of each newly generated spine at Day-3 was not significantly different between UE and VPA-exposed animals (mean ± s.e.m.; $P = 0.94$; Mann–Whitney U test; $n = 13$ and 10 dendrites from three UE and three VPA-exposed animals, respectively). **f** The survival rate of each pre-existing spine at Day-3 was not significantly different between UE and VPA-exposed animals (mean ± s.e.m.; $P = 0.091$). **g** The survival rates of newly generated spine at Day-3 were analyzed separately for the clustered and non-clustered spines ($P > 0.99$, Fisher's exact test; $n = 3$, 10, 3, and 7 for clustered GS, non-clustered GS, clustered GL, and non-clustered GL spines, respectively, in UE animals)($P > 0.99$, Fisher's exact test; $n = 14$, 8, 14, and 6, respectively, in VPA-exposed animals). \*\*$P < 0.01$; \*$P < 0.05$; n.s. not significant.

groups, the survival rates of newly generated spines (~50%) was considerably lower than that of pre-existing spines (~90%) (Fig. 3e, f). However, the survival rates of these two types of spines did not differ between the VPA-exposed and UE groups. Furthermore, spine survival rates in both the VPA-exposed and UE groups was independent of the presence or absence of clustering (Fig. 3g). The fact that spine survival rate was the same regardless of clustering suggests that interactions among newly formed spines have little effect on spine survival rate for some time after spine generation. In summary, clustered carryover spines comprised a much higher proportion of total spines in the VPA-exposed marmosets

than in the UE animals, while the survival rate of individual spines was equivalent in the two groups.

**Axonal boutons had a higher turnover rate in VPA-exposed marmosets than in controls**

Deficits in projection-specific connectivity or cortical interaction have often been discussed in ASD, as exemplified by local overconnectivity and long-range underconnectivity[40–42]. Spine turnover may depend on which neuron the coupled axons originate from[7]. We next analyzed marmoset axons that were transfected with colored fluorescent proteins that differed according to

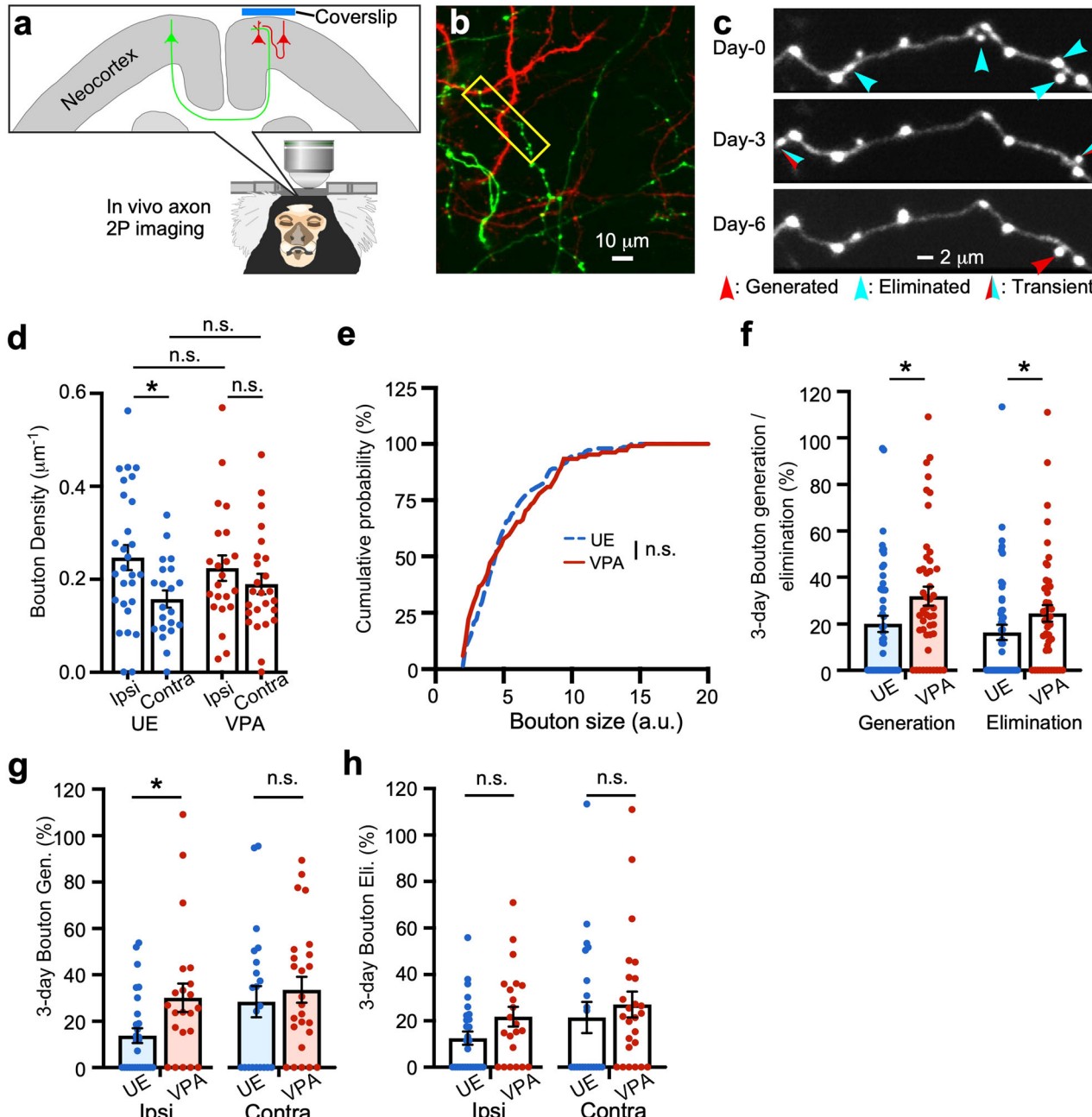

**Fig. 4 | A projection-specific elevation in synaptic turnover rate in VPA-exposed marmoset axons. a** AAV vectors expressing different colored fluorescent proteins were inoculated into each hemisphere of the dmPFC. **b** Axons from the contralateral hemisphere had green fluorescence, and axons from the ipsilateral hemisphere and dendrites had red fluorescence. **c** A representative axon from a VPA-exposed animal (the axon surrounded by the yellow rectangle in **b**) was imaged every 3 days. **d** Mean bouton densities are significantly larger in ipsilateral than in contralateral axons in UE animals (UE-ipsi vs. UE-contra, $P = 0.012$; $n = 28$ and 21 ipsilateral and contralateral axons, respectively, from three animals; two-way ANOVA followed by Fisher's LSD *post hoc* test). They are not significantly different between ipsilateral and contralateral axons in VPA-exposed animals (VPA-ipsi vs. VPA-contra, $P = 0.34$; $n = 22$ and 25 ipsilateral and contralateral axons, respectively, from two animals). The difference in bouton density between the ipsilateral axons or between

the contralateral axons was not statistically significant (UE-ipsi vs. VPA-ipsi, $P = 0.56$; UE-contra vs. VPA-contra, $P = 0.37$). **e** The bouton size distribution between these animals ($P = 0.56$; Kolmogorov–Smirnov test; $n = 137$ and 103 boutons in 49 and 47 axons in three and two UE and VPA-exposed animals, respectively). **f** There is a significant difference in three-day bouton generation rates between these animals ($P = 0.022$, Mann–Whitney U test; $n = 49$ and 47 axons in three and two UE and VPA-exposed animals, respectively), and also in 3-day bouton elimination ($P = 0.032$). **g, h** Three-day mean bouton generation rates (**g**) (mean ± s.e.m.; $P = 0.018$ and $P = 0.47$, Mann–Whitney U test; $n = 28$, 22, 21, and 25 for UE ipsilateral, VPA-exposed ipsilateral, UE contralateral, and VPA-exposed contralateral axons, respectively) and elimination rates (**h**) ($P = 0.098$ and $P = 0.28$) for each axon type. *$P < 0.05$; n.s. not significant.

the dmPFC hemisphere from which they projected (Fig. 4a). We expressed the red fluorescent protein tdTomato in neurons ipsilateral to the observation window in the PFC, and the green fluorescent protein mClover in neurons on the contralateral side (same axial position and same distance

from the midline; Fig. 4a). Therefore, we were able to observe red dendrites and red axons from local neurons, as well as green axons from contralateral neurons (Fig. 4b). In the magnified images, varicosities called presynaptic boutons were seen on the axons; new generation and elimination of these

varicosities were seen on each of the three observation days (Fig. 4c). Previous studies have shown that these boutons constitute presynapses[43], although some presynaptic terminals lack distinct varicosities (i.e., boutons) on the axons. In the present sample, most boutons were *en passant* boutons, and relatively few terminal boutons which had a neck between a varicosity and a parental axon shaft (Fig. 4c). In this study, we considered the boutons to represent the existence of presynapses. We first measured the axonal bouton density and found a significant difference between the mean values of the ipsilateral and contralateral UE marmoset axons. By contrast, the difference between the ipsilateral and contralateral axons in VPA-exposed marmosets was not significant (Fig. 4d). Next, we calculated the bouton size, which represents the synaptic weight[43], under the assumption that it was proportional to the maximum fluorescence intensity. We found that the distribution of bouton size was almost identical between the VPA-exposed and UE groups (Fig. 4e). The size distribution was consistent in both spines and boutons, indicating that the balance of excitatory neuronal inputs is maintained in this autism model. We then calculated the 3-day bouton generation and elimination rates. Consistent with the rates of spine generation and elimination, those of bouton generation and elimination were higher in VPA-exposed animals (Fig. 4f). Detailed analysis of each type of axon showed a significant difference in ipsilateral bouton gain between the two groups, whereas there was no significant difference in contralateral bouton gain (Fig. 4g) or loss (Fig. 4h). This suggested that the difference in the generation and elimination of synapses between the VPA-exposed and UE groups may have been caused by the alteration of ipsilateral (i.e., local) axons. The 3-day bouton turnover rate was higher than that of spines (spine: 10.2% and 12.5% for VPA-exposed group, 5.3% and 8.0% for UE group, for generation and elimination, respectively). These results suggest that a mechanism that regulates appropriate synaptic remodeling in a projection-specific manner exists in control animals and that it is disrupted (hyperdynamic turnover in the local circuits) in the ASD model.

## Oxytocin nasal administration modified dendritic spine clustering

Oxytocin, a prosocial neuropeptide, is a potentially effective treatment for ASD symptoms. Oxytocin modifies the social skills of typically developed human individuals, and improves sociality in mouse ASD models[44–46]; therefore, its potential as a therapeutic agent for ASD has long been anticipated. However, the benefits of oxytocin in human clinical trials are currently inconclusive[47–50]. In order to better understand the clinical applications of oxytocin, it is therefore important to investigate its mechanism of action on synaptic phenotypes, which were characterized in this study in VPA-exposed marmosets. Thus, we next examined how the aforementioned properties of cortical spines changed after the administration of oxytocin. Oxytocin receptors have been shown to be expressed in the mouse cerebral cortex presynapse and postsynapse of putative excitatory neurons, but are not yet known in marmosets[51–53]. We have demonstrated the presence of the oxytocin receptor protein through Western blot analysis within the synaptosomal fraction isolated from the gray matter of the adult marmoset prefrontal cortex (Supplementary Fig. 6). Marmosets were administered saline and then oxytocin, both intranasally (Fig. 5a). The average results from all marmosets failed to show a significant effect of oxytocin on spine generation and elimination rates (Fig. 5b; Supplementary Fig. 7a). Alternatively, analysis of the distance between generated spines suggested that oxytocin may affect the clustering efficiency of newly generated spines. Thus, we next examined the effect of oxytocin administration on spine clustering (Fig. 5c–j). After the oxytocin treatment period began, the proportion of clustered generated spines gradually decreased in VPA-exposed animals, while that of non-clustered generated spines increased, although the difference was not significant (Fig. 5c, d). We therefore tested whether the clustering bias was altered by the administration of oxytocin using simulations similar to those used in Fig. 2d–f. The clustering bias between generated spines observed in VPA-exposed animals during saline administration was reduced by nasal oxytocin administration (Fig. 5h–k). In contrast, the proportion of clustered generated spines gradually increased in

UE animals, disappearing statistically significant differences between UE and VPA-exposed animals before oxytocin administration (Fig. 5c). The percentage of non-clustered generated spines gradually decreased in UE animals, showing statistically significant difference in the D6–9 period (Fig. 5d). The clustering bias between generated spines observed during saline administration was increased in UE marmosets by nasal oxytocin administration (Fig. 5e–g, k). UE and VPA-exposed animals may be in different states with respect to synaptogenesis of the spines. The underlying mechanism remains elusive and warrants further investigation. Similarly, clustered spine elimination was enhanced by oxytocin administration in UE animals (Supplementary Fig. 7b). Conducting clustering bias analyses, however, we found oxytocin had a less pronounced effect on the proximity of clustering between eliminated spines compared to generated spines either in UE or VPA exposed animals. (Supplementary Fig. 7b, c–j).

It is known that there is sexual dimorphism in oxytocin receptor expression that well investigated in rodents[51,54,55]. On the other hand, it has been reported that an absence of sex differences in OT-immunoreactive neurons in marmoset brain regions such as paraventricularis and supraopticus of the hypothalamus as well as in the bed nucleus of the stria terminalis and the medial amygdala[56]. We therefore examined the effect of sexual differences on the effects of oxytocin by displaying scatter plots of the generation or elimination rate of spines before and after oxytocin administration (Supplementary Figs. 8–11). The scatterplot data for each dendrite shown separately for UE and VPA-exposed males and females. We could not detect statistically significant sex differences in VPA-exposed marmosets with respect to total, clustered, and non-clustered spine generation/elimination, potentially due to the considerable variation in values. Similarly, in the UE marmosets, although we detected statistically significant sex differences in several items (total and clustered spine generation at D6–9, non-clustered generation at D3–6, and total spine elimination at D0–3), we could not find any systematically comprehensible sex differences across the parameters under investigation.

We next analyzed the effect of oxytocin on axons. The bouton densities of each axon species were not significantly different before and after oxytocin administration in UE and VPA exposed marmosets (Supplementary Fig. 12a, b). The bouton generation/elimination of the ipsilateral axon, but not the contralateral axon, was significantly different between UE and VPA-exposed animals at D0–3, which is subsequently less pronounced at D3–6 and D6–9 (Supplementary Fig. 12c, d). These results suggest that oxytocin administration may have particular effects on the ipsilateral axons in the UE and VPA-exposed marmosets. Due to the lack of enough coincident events, we were unable to analyze the interactions between newly generated or eliminated axonal boutons.

## Gene expression analysis in adult ASD model marmosets

To explore the molecular mechanisms of altered spine dynamics in VPA-exposed marmosets and to assess the validity of adult VPA-exposed marmosets as a model of idiopathic ASD, we performed transcriptome analysis of the cerebral cortex using custom-made marmoset microarrays. Among the 9296 genes expressed in cortical tissues, there were 2484 differentially expressed genes (DEGs) with an adjusted *P*-value for multiple comparison ($P_{adj}$) of <0.05. First, we compared the DEGs in the cortical regions associated with social behavior between VPA-exposed marmosets and the postmortem brains of humans with ASD[57], and found a significant positive correlation between the two groups[58] (Fig. 6a; Supplementary Data 1). This was similar to our previously reported similarity between juvenile VPA-exposed marmosets and humans with ASD[22]. In addition, the modulation of gene expression in adult marmoset models and in human ASD was compared in each gene module that was previously configured by weighted gene co-expression network analysis (WGCNA) for samples from typically developed humans and those with ASD[22,58]. In both the marmoset model and human ASD, the majority of genes in modules associated with neurons and oligodendrocytes were downregulated (Fig. 6b), while genes in modules associated with astrocytes and microglia were upregulated (Fig. 6b). The direction of gene expression modulation of adult VPA-exposed marmosets

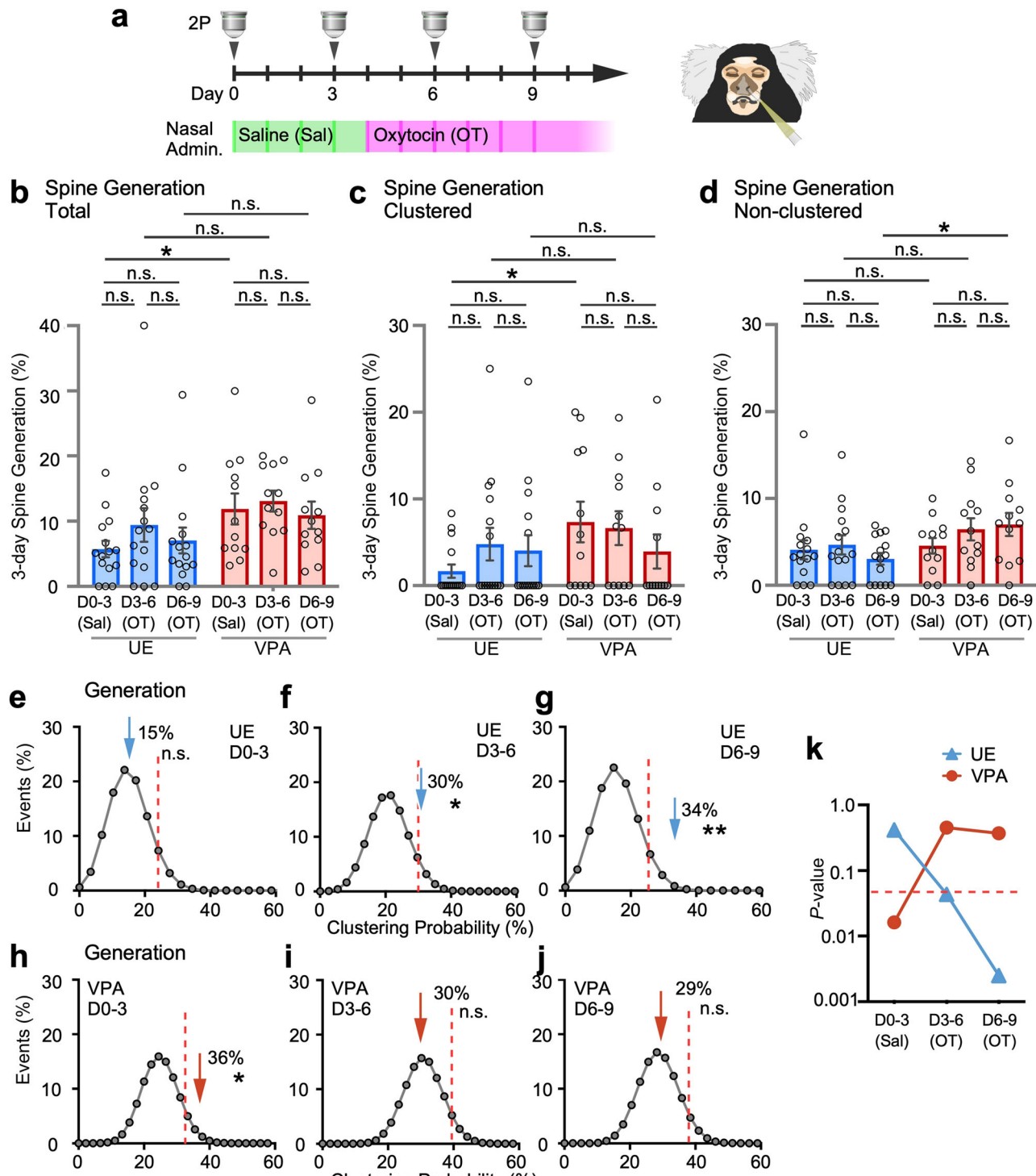

**Fig. 5 | Oxytocin modifies spine generation proximity in UE and VPA-exposed marmosets. a** Schematics of oxytocin nasal administration and two-photon (2P) imaging (Left) and transnasal administration using a micropipette (Right). Saline was given to marmosets in the green-shaded period, while oxytocin was given in the magenta-shaded period. **b** Three-day total spine generation rates (mean ± s.e.m.; Two-way ANOVA, group: $P = 0.011$, time: $P = 0.42$, interaction: $P = 0.80$; *post hoc* Tukey's test, D0–3 (UE vs. VPA): $P = 0.035$; $n = 15$ dendrites from four UE animals; $n = 12$ dendrites from three VPA-exposed animals). **c** Three-day clustered spine generation rates (mean ± s.e.m.; Two-way ANOVA, group: $P = 0.085$, time: $P = 0.63$, interaction: $P = 0.30$; *post hoc* Tukey's test, D0–3 (UE vs. VPA): $P = 0.038$; $n = 15$ dendrites from four UE animals; $n = 12$ dendrites from three VPA-exposed

animals). **d** Three-day non-clustered spine generation rates (mean ± s.e.m.; Two-way ANOVA, group: $P = 0.015$, time: $P = 0.51$, interaction: $P = 0.30$; *post hoc* Tukey's test, D6-9 (UE vs. VPA): $P = 0.017$; $n = 15$ dendrites from four UE animals; $n = 12$ dendrites from three VPA-exposed animals). **e–j** Simulation analysis to analyze the effects of oxytocin on clustering bias of newly generated spines in the UE (**e–g**) and VPA-exposed (**h–j**) animals ($n = 15$, 24, and 15 newly generated spine pairs from four UE animals, and $n = 32$, 32, and 28 pairs from three VPA-exposed animals, during the D0–3, D3–6, and D6–9 periods, respectively). Dotted red lines show 95th percentiles. **k** The P-values expressed in logarithm from (**e–j**) are indicated. **\*\***$P < 0.01$; **\***$P < 0.05$; n.s. not significant.

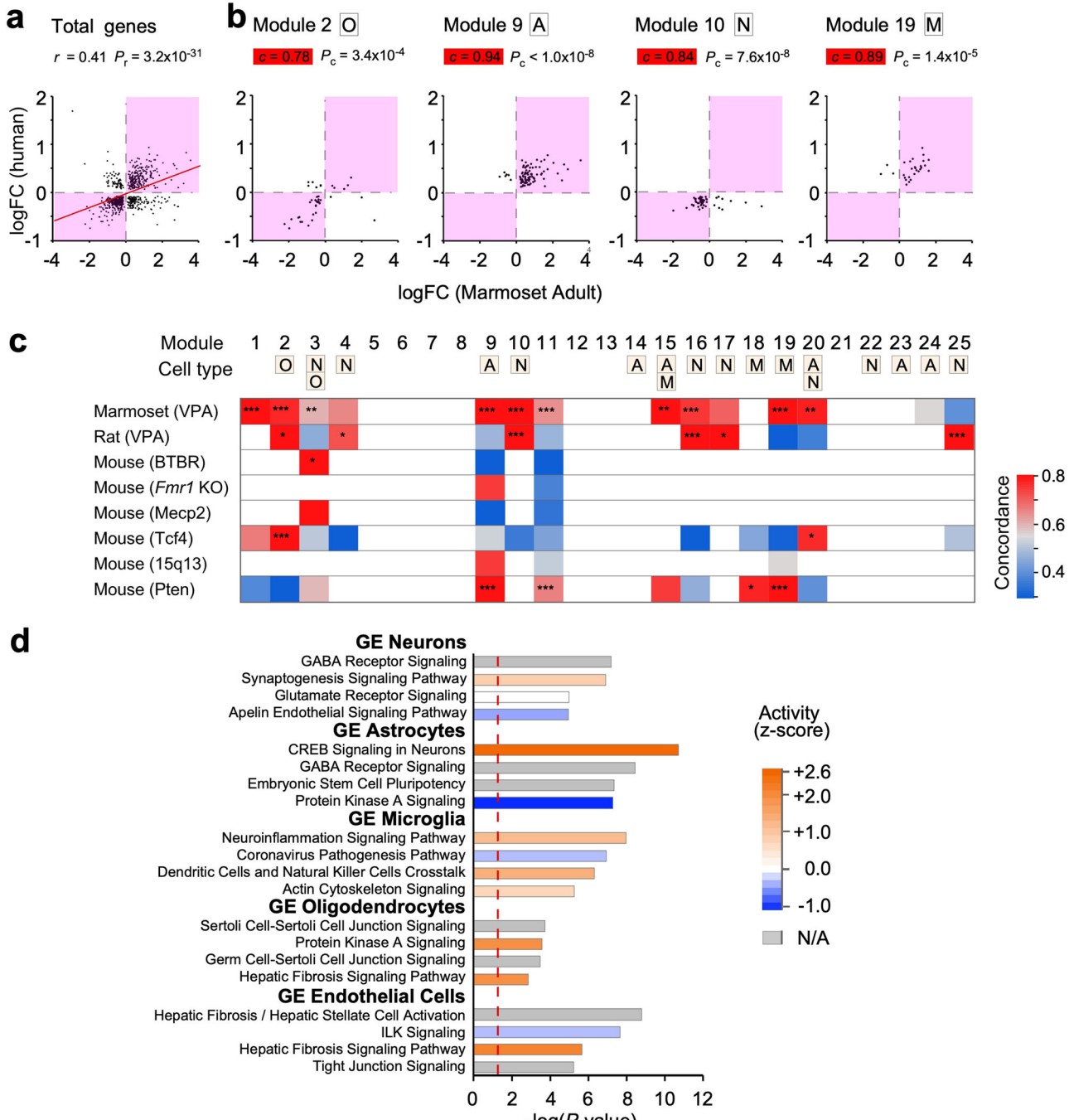

**Fig. 6 | Gene expression modulations in adult marmoset social neocortical areas reflect those in human ASD as determined by microarray analysis.**
**a, b** Relationship of gene expression modulation between adult marmosets and postmortem samples from humans with ASD. As a measure of gene expression modulation, log fold-change (logFC) values were computed between ASD model animals and control animals, and then compared with human ASD data. Modulated genes with $P_{adj} < 0.1$ are plotted. Pearson's correlation coefficient ($r$) and the P-value for the correlation coefficient ($P_r$) was shown in **a**. The magenta-shaded areas represent the first and third quadrants, and the concordance was calculated as the percentage of genes in that region. One-sided P-values for the concordance ($P_c$) are computed using a binomial distribution and are shown in **b**. The red line in **a** is the linear regression. **c** Concordance of gene expression modulations across human gene module 1–25 was examined in eight ASD model animals detailed in the Methods section. The degree of concordance in the model animals with those of human ASD

is shown in pseudo color for each gene module. The pseudo colors were displayed only when at least eight genes in the module were able to be analyzed in the human and animal models. Asterisks represent $P_c$: *$P < 0.05$, **$P < 0.01$, ***$P < 0.001$. For the rat VPA-exposed model and BTBR mice, genes with $P_{adj} < 0.05$ were selected; for other models and human ASD, those with $P_{adj} < 0.1$ were selected. The cell type(s) of the module (N: neuron, A: astrocyte, M: microglia, O: oligodendrocyte) indicate the cell category to which the genes of the module are most relevant (See Methods). **d** Gene enrichment (GE) for each cell type. The genes most closely associated with neurons, astrocytes, microglia, oligodendrocytes, and endothelial cells were analyzed. The color of the bars represents the direction of regulation of the pathway based on the logFC values. The P-values of enrichment were provided by the IPA software. The red dotted line represents the significance threshold ($P = 0.05$); N/A not available.

was very similar to that of human ASD samples in 10 modules, compared to nine modules for juvenile VPA-exposed animals[22] (Fig. 6c). The high similarity between VPA-exposed marmosets and human ASD contrasted with representative rodent monogenetic ASD models, which replicated human modulations only in limited gene modules and cell types (Fig. 6c). One explanation for this may be that these transgenic mice are a model of a specific type of ASD in which one gene contributes particularly strongly, which is not the case in idiopathic ASD.

We also analyzed DEGs in each cell type. This analysis confirmed the presence of a VPA-exposed marmoset–specific phenotype across the four cell types, as well as in vascular endothelial cells in the brain (Fig. 6d; Supplementary Data 2). Pathway analysis of cell type–specific DEGs showed pathway abnormalities in each of these cell types. For example, microglial modules exhibited abnormalities associated with inflammation, which fits well with the inflammation hypothesis of ASD[59].

## Discussion

In this study, we used in vivo two-photon microscopy to monitor neurite dynamics in the apical tuft dendrites of dmPFC layer-2/3 pyramidal neurons and adjacent axons in VPA-exposed marmosets. The main findings were as follows. First, new spines were generated in clusters and subsequently carried over much more frequently than non-clustered spines. Second, turnover rates of both dendritic spines and presynaptic boutons were higher than in control animals. Specifically, bouton turnover in VPA-exposed marmosets was significantly greater in local axons, but not in callosal axons, compared to controls, indicating possible maladjustment of projection-specific circuit plasticity. Finally, nasal administration of oxytocin to model marmosets reduced the tendency of spines to cluster without affecting their turnover rate in VPA-exposed marmosets. These spine-associated effects of oxytocin support its therapeutic potential in individuals with ASD.

The process by which coactivating spines were formed in close proximity to each other during operant tasks was studied using a combination of behavioral tasks and functional two-photon imaging[5]. Moreover, in layer-2/3 pyramidal neuron dendrites of the ferret visual cortex, neighboring spines tend to have similar visual representations[60]. These and other reports suggest a link between the proximity of the distance between generated spines and similarities in their reactivity. In fact, electron microscopy studies have shown that adjacent spines often serve as postsynaptic targets for single axons[61]. There are several studies that have shown that synaptic plasticity diffuses from stimulated spines into neighboring spines and interacts there[5,32,37,62–65]. However, the relationship between heterosynaptic plasticity and the subsequent generation or elimination of surrounding spines is not yet fully understood and is a subject for future research. The observed higher spine turnover, or more precisely, the larger population of spines with heightened turnover in the VPA-exposed animals, may suggest the potential for the formation of more efficient neuronal circuits[66]. Clustered generated spines may also help stabilize formed circuits and accelerate learning due to their redundant involvement in the circuit[67]. In fact, CCR5 KO and MeCP2 duplication mice commonly show elevated spine turnover and clustering as well as high learning efficiency[10,18]. On the other hand, over-efficient formation of neural circuits due to excessive synaptic clustering may result in the preservation of synapses that should be removed and may prevent the formation of additional neural circuits. Consistent with this hypothesis, VPA-exposed marmosets learned faster than UE marmosets during the first discrimination phase of a spatial reversal learning task, but learned slower after paradigm reversal[68]. Conditions that constantly enhance spine clustering could potentially modify the efficiency of neural circuit reconnection, consequently impacting social learning processes.

Previous in vivo two-photon microscopy studies of ASD model mice have shown a common phenotype of high spine turnover in the motor and early sensory cortices that are less associated with ASD core symptoms than in the PFC[7–10]. Accumulating evidence, including the results of this study, suggests that ASD-related genes and environmental factors converge across species, resulting in a phenotype of increased spine dynamics that

continuously promotes neural circuit remodeling. Compared to spine turnover (generation and elimination), spine clustering in ASD animal models has been reported very infrequently to date, and only in the motor and visual cortices in MeCP2 duplication models[10].

The synaptic connections of local neural circuits exhibited more plasticity in VPA-exposed animals than in UE animals, whereas the long-distance callosal neural circuits did not show plasticity differences. This suggests that projection-specific plasticity control may be altered in ASD[40,41,69]. The excessive plasticity in the local neuronal circuits of VPA-exposed marmosets is reminiscent of the enhanced long-term potentiation of layer-5 pyramidal neurons in acute slice preparations in a VPA-exposed rat ASD model[70]. However, differences in synaptic remodeling of different types of cortical inputs have never been compared in living animals or in vitro in ASD models. Postsynaptic neuron–specific deletion of the Fmr1 gene in mice reduced the amplitude of synaptic transmission in callosal inputs but not in local inputs, suggesting the existence of differential regulation between the group of inputs[71]. Several experiments, including single-cell transcriptome analyses, have shown a variable number of DEGs between cortical layer–specific excitatory neurons[72] or between cortical areas[57,73] in human ASD. These differences may serve as the basis for projection-specific circuit plasticity. Abnormalities in brain connectivity strength between ASD-relevant brain regions in humans with ASD have been demonstrated by functional brain imaging techniques such as functional magnetic resonance imaging (fMRI) and magnetoencephalography (MEG) /electroencephalography (EEG), using the degree of brain activity coherence as an indicator[1,41,43]. Differences in the degree of bouton dynamics between local and callosal connections in VPA-exposed marmosets suggest that in addition to variations in connection strength, changes in projection-specific synaptic remodeling may also contribute to ASD symptoms.

To identify therapeutic targets, it will be valuable to determine the molecular mechanisms that govern spine turnover and spine clustering in VPA-exposed marmosets[74]. Our study demonstrated that oxytocin reduced clustering of generated spines in ASD model marmosets without altering spine turnover, while in mice the effect of oxytocin on synaptic plasticity has been studied in various brain regions such as the olfactory system, hypothalamus, and neocortex[45,75]. In the present study, we have substantiated the presence of the oxytocin receptor protein through Western blot analysis within the synaptosomal fraction isolated from the gray matter of the adult marmoset prefrontal cortex (Supplementary Fig. 6). This observation implies that chronic oxytocin exposure potentially exerts a direct inhibitory influence on spine clustering by modulating intracellular signaling pathways, notably the MAPK/ERK pathway, downstream of oxytocin receptor activation. Differences in oxytocin systems between rodents and primates may considerably affect therapeutic efficiency, which underscores the importance of primates in translational research on oxytocin[53]. Spine clustering in the MeCP2 duplication mouse model was suppressed by an inhibitor of Ras–MAPK(ERK) signaling without altering the spine generation rate[76]. Ras, like other proteins such as Rho, Rac, and cofilin, has been reported to diffuse from activated spines to neighboring spines via dendrites and is thought to alter the plasticity of neighboring spines[5,33,34,77]. Interestingly, transcriptome analysis of our model marmosets showed four times greater upregulation of TGF-β2, which is upstream of the Ras–MAPK pathway, than control marmosets. Synaptic plasticity involves not only neurons but also diverse glial cells. We found that microglia demonstrated activation of signals related to viral infection, inflammation, and reorganization of the actin cytoskeleton (Fig. 6d). On the other hand, it has been shown that the stress hormone corticosterone increases spine turnover in mice[78]. Indeed, as in mouse ASD strains[79] and humans with ASD[80], our ASD model marmosets have aberrant cortisol responses[81], suggesting that environmental factors such as anxiety and endocrine system disorders may be responsible for the increase in synaptic remodeling. Therefore, oxytocin, which reduces spine clustering, could be beneficial for individuals with ASD when combined with the suppression of stress responses.

## Limitations of the study

In this study we did not investigate inhibitory synapses. The location of inhibitory inputs and the local balance between excitatory and inhibitory inputs may also have a considerable effect on nonlinear information integration by dendrites. We were also unable to analyze the genetic, anatomical, or electrophysiological characteristics of the pyramidal neurons that we examined. We could not conclude a systematic sex difference in the present data (depicted in Supplementary Figs. 8–11 for oxytocin treatment). Activity-dependent labeling of neurons would allow for better characterization of the relationship between neuronal activity and spine clustering. Electron microscopy analysis of clustered generated spines would have made it possible to confirm the presence of common presynaptic partners. We believe that the future implementation of these approaches will better clarify the nature of spine clustering in autism models.

## Inclusion and diversity

We support inclusive, diverse, and equitable research conduct.

# Methods

### Animals

All experimental procedures were approved by the Animal Research Committee of the National Center of Neurology and Psychiatry. We have complied with all relevant ethical regulations for animal use. Common marmosets (*Callithrix jacchus*) were housed in captivity at the National Center of Neurology and Psychiatry under a 12-h/12-h light/dark cycle and were fed food (CMS-1; Clare Japan) and water ad libitum. Temperature was maintained at 27–30 °C and humidity at 40–50%.

### Marmoset ASD model

Serum progesterone levels in female marmosets were regularly measured to determine the date of fertilization. Four percent VPA dissolved in 10% glucose solution was administered intragastrically to pregnant marmosets at 200 mg/kg/day daily for 7 days starting on gestation day 60. No doses were given to control mothers. The obtained ASD model offspring from VPA-administered mothers and control offspring from UE mothers were kept in their home cages until the day of inoculation with AAV vector. We used four UE marmosets (monkeys KR (f, 1.3), TR (f, 3.2), VR (m, 2.2), and BH (m, 1.3)) and three VPA-exposed marmosets (monkeys PR (f, 1.5), SH (f, 1.4), and MS (m, 1.3)); f or m and numbers in parentheses indicate sex and age in years at the time of AAV inoculation, respectively.

### AAV vector inoculation of the marmoset neocortex

Each experimental marmoset was pretreated with the antibiotic cefovecin sodium (8 mg/kg body weight, intramuscular (i.m.); Zoetis), prednisolone (2 mg/kg body weight; i.m.), the analgesic ketoprofen (2 mg/kg body weight, i.m.), and atropine (0.15 mg/kg body weight, i.m.). It was then anesthetized with ketamine (15 mg/kg body weight, i.m.; Daiichi-Sankyo) and xylazine (1.2 mg/kg body weight, i.m.; Bayer), which were supplemented with sevoflurane inhalation (2–4%). The marmoset was then fixed to a stereotaxic instrument (SR-6C-HT; Narishige, Tokyo, Japan). During all surgical procedures, marmosets were also supplied with humidified oxygen as needed, and warmed to 37–39 °C with a heating pad (FST-HPS; Fine Science Tools, North Vancouver, Canada). Body temperature, SpO$_2$, heart rate, cardiac electrogram, respiratory rate, and actual concentration of oxygen and sevoflurane were measured using a biomonitor (BSM-5132; Nihonkoden, Tokyo). After incision of the skin at the midline, the skull was exposed and a 1-mm-diameter hole was made bilaterally above the PFC Brodmann area 8 (11.5 mm anterior to the interaural line, 3 mm lateral to the midline) using a dental drill.

A puller (PC-100; Narishige) was used to create a micropipette (tip diameter 30–50 μm) from a micro glass tube (1.0-mm diameter; TW100F-4; WPI, USA). The pulled micropipette was then beveled using a micro grinder (EG-402; Narishige) and was sterilized by overnight exposure to UV light or ethylene oxide. The pipette was set to a glass microsyringe (Model 1701,10 μL; Hamilton, USA) using a Priming kit (55750-01; Hamilton), and

the microsyringe was back-filled with silicon oil and attached to a microsyringe pump (LEGATO 130; WPI) fixed to the manipulator of a stereotaxic instrument (SM-15R; Narishige). The micropipette was then tip-filled with AAV vector suspension and slowly inserted through the dura and into the neocortex, avoiding large blood vessels. The total 0.5-μL volume of AAV vector was then inoculated at 0.1 μL/min at a depth of 0.8 mm from the surface. The viral preparations were adjusted to the following final concentrations: $1 \times 10^9$ vg/mL for rAAV1-Thy1S-tTA and $1 \times 10^{12}$ vg/mL for AAV1-TRE3-tdTomato, expressing red fluorescence protein; $5 \times 10^9$ vg/mL for rAAV2/1-Thy1S-tTA and $1 \times 10^{12}$ vg/mL for rAAV2/1-TRE3-mClover, expressing green fluorescence protein. After the 5-min incubation period, the micropipette was retracted, a silicon plug was stuffed into the cranial hole, and the skin incision was closed with sutures. After recovery from anesthesia, the marmoset was returned to its home cage.

### Imaging window preparation

Each experimental marmoset was pretreated with the antibiotic cefovecin sodium, prednisolone, the analgesic ketoprofen, and atropine. It was then anesthetized with ketamine and xylazine, which were supplemented with sevoflurane inhalation (2–4%). The marmoset was then fixed to a stereotaxic instrument. The marmoset was also supplied with humidified oxygen and warmed up to 37–39 °C. After skin incision at the midline, the skull was exposed and a 4-mm-diameter hole was made at the point where the AAV was inoculated using a dental drill fixed to a stereotaxic manipulator (SM-15R; Narishige). The dura mater was carefully removed using microsurgical ophthalmic scissors, fine forceps, and a microhook to minimize any pressure applied to the surface of the brain. A piece of coverslip, consisting of three laminated circular coverslips (5-mm diameter + 3-mm diameter × 2), was used to cover the exposed brain surface[28]. The coverslip was fixed using dental acrylic (Fuji-Lute; GC Corp., Tokyo, Japan) to create an imaging window in the skull. A custom-made stainless-steel recording chamber (ICM, Tsukuba, Japan) was then attached to the skull using dental resin (Bistite II; Tokuyama Dental, Japan) such that the imaging window was in the center of the chamber. After the imaging window was covered with an acrylic lid, the marmoset was allowed to recover from the anesthesia in the monkey ICU room and was then returned to its home cage.

### In vivo two-photon excitation microscopy

In vivo two-photon imaging of neurites was performed using an upright microscope (BX61WI; Olympus) equipped with a laser scanning microscope system (FV1000; Olympus) and a water-immersion objective lens (XLPlanN, 25×, NA 1.05; Olympus). The system included mode-locked, femtosecond-pulse Ti:sapphire lasers (MaiTai; Spectra Physics) set at a wavelength of 980 nm.

Imaging sessions were conducted every 3 days. A marmoset was pretreated with atropine and anesthetized with sevoflurane inhalation (3–5%). The marmoset was laid in the prone position and its head was fixed using the imaging chamber and brass posts (Fig. 1a). In each case, the imaging location was identified based on blood vessel morphology. Tuft dendritic branches (<100 μm depth) of layer 2/3 pyramidal neurons were used for two-photon imaging experiments. An objective lens correction collar was manually set just before imaging so as to minimize spherical aberrations (i.e., to acquire the brightest image possible). The reciprocal scan mode was used to scan each *xy*-image (256 by 256 pixels; 65 msec/frame). Three-dimensional fluorescent images with 51 *xy*-images, each separated by 0.5 μm, were obtained at each imaging site.

### Dendritic spine and axonal bouton analysis

We applied a 2-pixel spatial Gaussian filter and then registered the obtained *xyz*-images using the StackReg or Multi-StackReg plug-in function of Image-J (Fiji)[82]. Spines were identified as protrusions from dendrites with an apparent head structure. Filopodial protrusions (which have no apparent spine head) were excluded from the analysis. Two spines observed during different sessions were considered to represent the same spine if the initial position subsequently exhibited the same distances from adjacent

landmarks. A spine was interpreted as lost if it temporarily disappeared. The minimum spine length was set at 0.4 μm. Snowman-shaped spines (shaped like two spheres stuck together) were interpreted as two separate spines. Only spines that appeared laterally were included in the analysis (this underestimated the spine density). Dendrites that were visible in all sessions were used in the analysis.

We also analyzed the axons that appeared in the same images as the dendrites. For axon analysis, we used five marmosets that expressed mClover in the right hemisphere and tdTomato in the left hemisphere (UE monkeys: TR, VR, BH; VPA-exposed monkeys: SH, MS). We applied a 2-pixel spatial Gaussian filter to the *xyz*-images and obtained *z*-substacked average images for a predetermined z number for each axon. By combining images obtained at the optimal depths, we created montage images for each axon, avoiding dendrites and other axons present at other imaging depths. We then examined the brightness along the axon shaft using the Plot-profile function of Image-J[82]. The ratio of bouton brightness to neighboring axon shaft brightness was then calculated. We did not see axons that were much thicker than the diffraction limit of our two-photon imaging, and discarded areas that overlapped with other dendrites and axons. Boutons were detected as bright swellings along axons, with intensity values at least two-fold higher than that of the flanking axon backbone[83]. Since we could not detect axonal synapses below the bouton threshold, the subthreshold boutons may have been misidentified as newly generated boutons when they enlarged and therefore exceeded the threshold in the next imaging session. As a consequence, the number of newly generated boutons may have been overestimated. Likewise, the number of boutons that were eliminated may also have been overestimated. This may explain the result that the overall bouton turnover was higher than the spine turnover. The 3-day bouton generation and elimination for unit axon length were obtained by dividing the number of boutons generated and eliminated over a 3-day period by the axon length. The 3-day bouton generation and elimination as a percentage were calculated by dividing the 3-day bouton generation and elimination for unit axon length by the mean bouton density. We did not exclude varicosities of transfer vesicles. In the next session, boutons located within 1 μm of the expected point were interpreted as identical to those in the previous session.

The percentage of newly formed spines was calculated as the number present at time point 2 but not at time point 1, divided by the total number present at time point 1 and multiplied by 100. The percentage of eliminated spines was calculated as the number present at time point 1 but not at time point 2, divided by the total number present at time point 1 and multiplied by 100.

## Monte Carlo simulation of newly generated or eliminated dendritic spines

To obtain unbiased distributions of the inter-spine distance of newly generated or eliminated spines, each dendrite was concatenated in random order, producing one long dendrite repeatedly (1000 times permutation; Fig. 2d and Supplementary Fig. 3). The average clustering probability was then calculated. We set the clustering threshold at 3 μm since several biochemical, physiological, and structural studies have suggested that a 3–10 μm distance between spines facilitates sharing of resources, spine co-activation, and learning-induced structural plasticity[32–38]. In Supplementary Fig. 2a, the threshold applied for the analyses (3 μm) was confirmed to be appropriate because it is located at the lowest end of the plateau[10]. Monte Carlo simulation was performed to determine whether a clustering bias existed in the process of spine generation or elimination[18] (Supplementary Fig. 2c–p). The simulation used the total dendrite length and the total number of generated and eliminated spines. The positions of generated and eliminated spines on the dendrites were then regenerated with a uniform random distribution using Excel software. The distances between generated and eliminated spines were computed for all dendrites and the percentage of clustered events was calculated (Fig. 2d). In Supplementary Fig. 2o, p, actual clustering probabilities were plotted against simulated clustering probabilities (chance level). The contact point in Supplementary Fig. 2o between

the plot of VPA and the black dotted line parallel to the diagonal line provides Youden's index which provides optimal cut-off point of two distributions[39]. This process again confirmed that 3 μm provides optimal cut-off value for removing random clustering influence. As clustered spines may appear more on specific dendrites, another simulation was conducted on individual dendrites without connecting each dendrite, and similar results were obtained (Supplementary Fig. 5b–e). The above simulations were performed with 100,000 iterations to obtain the distribution of clustering events, and the results are shown in Figs. 2e, f, 5e–j, and Supplementary Figs. 2c–n, 3d, e, 7d–i by connected gray lines.

## Dendritic spine volume measurement

We calculated the spine-head volume by partially summing the fluorescence values of five sequential *z*-images by taking the moving average of the image stack along the *z*-plane. This was done to avoid summing other dendrites or axonal fibers present in different imaging planes. Because the thickness of dendritic spines is near the diffraction limit of a two-photon microscope, partially summed values (2 μm range in the *z*-direction) can be used to reflect spine volumes. Thus, the maximum value of *z*-moving average images allowed us to obtain good approximations of the total *z*-summed stacked images. The normalized spine volume was obtained by dividing the spine fluorescence intensity by the fluorescence intensity of the adjacent dendrite shaft.

## Nasal oxytocin administration to VPA-exposed marmosets

New World monkey oxytocin (Pro-8 oxytocin[84]) was custom synthesized (GenScript, Piscataway, NJ, USA) and was dissolved in 0.5 mg/mL of 0.9% NaCl solution. The oxytocin solution was typically administered 2.5 h before the two-photon imaging and at approximately same time on days without two-photon imaging. Oxytocin solution was applied to the nasal cavities of each marmoset at a dose of 150 μg/kg body weight as in a previous study[84], using a micropipette equipped with a soft silicon tube on the tip. Oxytocin was administered every day (monkey VR, BH, TR and MS) or every other day (monkeys KR, PR and SH) starting on day−4 (Fig. 5a). Verification of oxytocin delivery to the cerebral interstitial fluid was verified by measuring the plasma Pro8-oxytocin concentration using a sandwich competitive chemiluminescent enzyme immunoassay (CLEIA) (ASKA Pharma Medical, Fujisawa, Japan). The Pro8-oxytocin concentration in all blood samples was monotonically increased 30 min after nasal oxytocin administration (mean ± s.d. = 30.6 ± 14.8 pg/mL; $n = 2$ males and 3 females) but not 30 min after nasal saline administration (mean ± s.d. = 9.9 ± 1.7 pg/mL; $n = 2$ males and 3 females); thus, we concluded that Pro8-oxytocin was successfully delivered to the interstitial fluid of the neocortex.

## Microarray gene expression analysis

Microarray analysis of marmoset gene expression was performed using brain samples from three adult VPA-exposed and three UE marmosets, as previously described[22]. To increase data reliability, we pooled the data from the social brain areas (area 12 and area TE) that are affected in humans with ASD. Tissue sampling and GeneChip analysis were conducted as previously reported[22]. Briefly, marmosets were anesthetized with ketamine hydrochloride (50 mg/kg, i.m.) and sodium pentobarbital (90–230 mg/kg, intraperitoneal; Kyoritsu Seiyaku). After reflections had completely disappeared, the animals were transcardially perfused with diethyl pyrocarbonate-treated phosphate-buffered saline, and the cortical tissue was isolated and immersed in RNAlater (Thermo Fisher Scientific). Total RNA was extracted using the RNeasy Mini Kit (Qiagen, Hilden, Germany). RNA integrity was assessed using a Bioanalyzer (Agilent Technologies), and samples with RNA integrity number values > 7 were evaluated. Biotin-labeled cRNA probes were prepared using the GeneChip 3′IVT Express Kit (Affymetrix). The probes were hybridized to a custom-made microarray (Marmo2a520631F) using the GeneChip Hybridization, Wash, and Stain Kit (Affymetrix). Microarrays were scanned using a GeneChip Scanner 3000 (Affymetrix) and processed using MAS5, and the reliability of probe detection was examined. Data were normalized using GCRMA. Genes with log2 expression values greater than

5 were considered to be expressed in brain tissue. The log fold-change (logFC) value was used as a measure of gene expression modulation and evaluated using Welch's $t$ test with Benjamini–Hochberg adjustment ($P_{adj}$). For affected genes with multiple probes, the data from the probe with the lowest $P_{adj}$ was used. To compare gene expression modulations in the marmoset model with those in human ASD postmortem samples[57] commonly modulated genes with $P_{adj} < 0.1$ were used.

## Comparison of log fold change (logFC) values between human ASD and animal models, as determined by gene co-expression modules

To compare gene expression changes between the ASD group and the typically developed human group, or between the ASD model animals and controls, logFC values were calculated for the genes analyzed by the gene chip. The concordance values were then calculated for each gene module[57,58] to show consistency in the direction of gene expression changes between human ASD and the animal models (Fig. 6c). The concordance value was the percentage of genes that changed in a common direction (first and third quadrants in Fig. 6b) in human ASD and the model animal. For modules with at least eight common genes between the animal model and human ASD, the concordance values are shown in color in Fig. 6c. The logFC values of VPA rats[85] (age 35 days), BTBR mice[86] (age 4 months), *Fmr1* KO mice[87] (age 8–14 weeks), *MeCP2* heterozygous KO mice[88] (female, age 5 weeks), *Tcf4*tr/+ mice[89] (age 60–80 days), 15q13.3 deletion mice[90] (age 10–22 weeks), and *Pten*m3m4 mice[91] (age 6 weeks) were obtained from previously published studies. To compare gene expression modulations in the marmoset model with those in postmortem samples of humans with ASD[57], commonly modulated genes with $P_{adj} < 0.1$ were used. For the rat VPA models and BTBR mice, genes with $P_{adj} < 0.05$ were selected; for other models and human ASD, genes with $P_{adj} < 0.1$ were selected. To compare gene expression modulations in the rodent models with those in human ASD, gene symbols were converted using HomoloGene (NCBI, https://www.ncbi.nlm.nih.gov/homologene, release 68). Due to the small number of genes that are commonly affected, mouse models of maternal immune activation and *Shank3* knockout were not included in the list.

## Signaling pathway estimation of marmoset genes

Pathway analysis of marmoset genes was conducted using IPA software (Qiagen, Summer Release 2020). Genes specific to brain cell types (neurons, astrocytes, microglia, oligodendrocytes, and endothelial cells) were selected from genes with mean logFC gene enrichment values > 2, among the top-ranked cell type–enriched genes based on the human single-cell analysis data in a previous report[92].

## Examination of glial cell activation

Marmoset brains fixed in 4% PFA were replaced with 30% sucrose in 0.1 M phosphate buffer (pH 7.4). Frozen sections with 50 μm thickness were prepared using a microtome (Retoratome REM-710; Yamato-Kohki, Saitama, Japan). Brain slices were treated in antigen-retrieval solution (HistoVT One; Nacalai Tesque, Kyoto, Japan) at 70 °C for 20 min, followed by PBS washing and blocking (blocking buffer: 5% normal goat serum, 0.4% TritonX-100 in PBS) at room temperature for 60 min. After blocking, free-floating brain slices were incubated in a solution of anti-Iba1 antibody (1:2000; Fujifilm-Wako, #019-19741) or anti-GFAP antibody (1:10,000; Agilent-Dako, #Z0334) at 4 °C overnight. The slices were washed with PBS (10 min, three times) and incubated with biotin-labeled goat anti-rabbit secondary antibody (1:500; Vector Labs) at room temperature for 90 min. The slices were washed with PBS (10 min, three times), then reacted with the ABC complex for 90 min at room temperature (Vectastain Elite ABC kit; Vector Labs). After washing with phosphate buffer (10 min, three times), the slices were mixed with nickel-sensitized DAB reaction solution (0.01% 3,3'-diaminobenzidine-4HCl) to develop color. The samples were washed with phosphate buffer, mounted on glass slides, and dehydrated with ethanol and xylene. The glass slides were then sealed with coverslips for microscopic observation. Microglia and astrocytes were observed and cell numbers were counted using a Neurolucida 360 system equipped with an inverted microscope (MBF Bioscience, Williston, VT, USA).

## Western blotting

Three adult UE marmosets (female, 2.8 year-old; male, 6.8 y; female, 8.8 y) were deeply anesthetized with pentobarbital after ketamine preanesthesia. After confirming the disappearance of reflexes, ice-cold PBS was transcardially perfused and subsequent cardiac arrest was confirmed. While continuing ice-cold PBS perfusion, a portion of the skull was removed and the entire brain was harvested. The gray matter of the prefrontal cortex was excised from the brain and placed in a 2-mL plastic tube, which was immediately frozen on dry ice and stored at −80 °C until use. 53–93 mg of frozen prefrontal gray matter was placed in a Lysing Matrix D 2-mL tube (MP-Biomedicals, Irvine, CA, USA) cooled on ice, 800 μl of Syn-PER™ Synaptic Protein Extraction Reagent (Thermo Fisher Scientific, Waltham, MA, USA) and cOmplete mini Protease Inhibitor Cocktails (Merck, Darmstadt, Germany) was added. The tissue was homogenized by shaking the Matrix D tube at 3200 rpm for 30 s using a bead beater homogenizer (μT-12, Taitec, Koshigaya, Japan). The homogenate was then centrifuged at $1200 \times g$ for 10 min. The supernatant was further centrifuged at $15,000 \times g$ for 20 min to obtain a synaptosome fraction. The synaptosome fraction was re-suspended in 700 μl of SynPER and the protein components of the synaptosome were extracted by mixing 70 μl of 100% Trichloroacetic acid and allowing it to stand for 30 min on ice. The protein extract was collected by a centrifugation at $15,000 \times g$ for 5 min. SDS-PAGE sample buffer was then added to the extract and neutralized by adding 1 M Tris. SDS-PAGE (7.5% gel, 10 or 1 μg total protein per lane, for anti-oxytocin-receptor or anti-PSD95/anti-β-actin, respectively) was performed and proteins were transferred to PVDF membranes and stained with 0.2% Ponceau-S in 5% acetic acid. The membrane was then incubated in a blocking solution containing 5% skimmed milk, 0.2% Tween20, and 0.02% NaN₃ in PBS at 4 °C, overnight. Membranes were incubated with the following primary antibodies: anti-PSD95, mouse, Novus Biologicals, #NB300-556, 1:4,000; anti-Oxytocin receptor, rabbit, Abcam, #217212, 1:100; anti-β-actin, mouse, Santa Cruz, #sc-47778, 1:4,000 in PBS-T (0.1% Tween20 in PBS) for 3.5 h at room temperature. After twice of washing with PBS-T, the membranes were treated with secondary antibodies: Sheep anti-mouse IgG +HRP, Cytiva, #NIF825, 1:4,000; Goat anti-rabbit IgG +HRP, MBL, #458, 1:5,000 for 30 min at room temperature. After 4 washes with PBS-T, the blots were visualized with ECL Western Blotting Reagents, Cytiva, #RPN2109, and detected using Fusion-S solo imager (Vilber, Collégien, France).

## Statistics and reproducibility

All data are presented as mean ± s.e.m ($n$ = dendrite or axon numbers) unless otherwise stated. The Mann–Whitney rank sum test was used to analyze the data shown in Figs. 1c, d, 2b, 3e, f, and 4f–h, and Supplementary Figs. 1, 3b, 8–11. No adjustment of $P$-values for multiple comparisons was made for Supplementary Figs. 8–11. The difference in probability distribution between cumulative plots was calculated using the Kolmogorov–Smirnov test, as shown in Figs. 1e, f, 2c, and 4e, and Supplementary Figs. 3c, 4b, c. The differences between means were analyzed using the Kruskal–Wallis test in Supplementary Fig. 12a, b. The two-way ANOVA was used to analyze the data shown in Figs. 4d and 5b–d, and Supplementary Figs. 7a–c, 12c, d. In the Two-way ANOVA in Supplementary Fig. 12c, d, a mixed-effects model was used to complement missing values. The Fisher's exact test was used to examine the frequency of the carryover spines and that of the total spines in Fig. 3c, d and to examine the spine survival rates for each condition in Figs. 1g and 3g. The $P$-values in Fig. 5k and Supplementary Fig. 7j were obtained from the distribution of the simulation. The $r$ and $P$-values in Fig. 6a show Pearson's correlation coefficients ($r$) and their $P$-values ($P_r$) against zero. The concordance values ($c$) and their $P$-values ($P_c$) in Fig. 6b show the number of genes exhibiting concordant changes between the animal model and human ASD divided by the total number of genes, and the probability was calculated using the one-sided binomial test. Other statistical tests are specified in the text or figure legends or elsewhere in the

**Article**

Methods section. Quantification, simulation, and statistical analysis were performed using Microsoft Excel, GraphPad Prism 9.3.1., and Real Statistics software (for Kolmogorov–Smirnov test; www.real-statistics.com.). Analyses on spines were first conducted by an analyst blinded to the sample labels and subsequently validated by an independent analyst. In the context of axonal analysis, the quantification of fluorescence intensity was carried out in a semi-automated manner, delineating a linear region of interest (ROI) along the axon, utilizing Image-J (FIJI) software.

## Data availability

The microarray data generated in this study have been deposited at NCBI GEO under accession number GSE199560. All other data supporting the findings related to Fig. 6 are in Supplementary Data 1, 2. The source data behind the graphs in the paper can be found in Supplementary Data 3. Any additional information required to reanalyze the data reported in this paper is available from the corresponding authors on reasonable request.

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

## Acknowledgements

We thank H. Kasai of The University of Tokyo for helpful discussion and suggestions about two-photon imaging, and both H. Yamasue of Hamamatsu University School of Medicine and T. Minamimoto of QST Japan for their careful reading and thoughtful comments on the manuscript. We are also grateful to T. Araki, S. Wakatsuki and M. Iwasaki of the National Center of Neurology and Psychiatry (NCNP) for helpful discussion about experiments and marmoset surgery. We also thank N. Hasegawa, A. Tsuchiya, R. Shiraishi, M. Nakamura, T. Sato, W. Suzuki, A. Mishima, K. Mimura, and N. Miyakawa of our lab for supporting primate experiments, and R. Saito, Y. Katakai, and other members of the NCNP primate facility for caring for VPA-exposed and UE marmosets. A. Sawatari and S. Ikeda of Iwate University assisted with spine volume analysis. English editing was performed by Zenis, co. Ltd., Kyoto, Japan. This work was supported by Intramural Research Grants for Neurological and Psychiatric Disorders from the NCNP (3-5, J.N.; 29-6, N.I.), a Novartis Research Grant 2019 (S.W.), JSPS KAKENHI Grant Numbers JP18K06497, JP22K07363 (J.N.), and JP21K07535 (S.W.) and AMED Grant Number JP21dm0207066 (N.I.) and JP15dm0207001 (T.Y.).

## Author contributions

J.N. and N.I. designed the study; J.N., S.W., R.I., and K.Sak. performed the in vivo two-photon imaging; K.N. generated the autism model marmosets; J.N. and R.I. conducted image analyses; E.S., H.T., H.M., A.W., and T.Y. prepared the AAV vectors.; S.W., T.O., K.Su., K.H., K.Sai., and I.M. conducted the microarray analysis.; K.Sak. and R.I. performed immunohistochemistry; and J.N. and N.I. wrote the manuscript. All authors approved the final version of the manuscript. Correspondence and requests for data should be addressed to J.N. or N.I.

## Competing interests

The authors declare no competing interests.
