## [Peer Review File · Communications Biology]

Reviewers' comments:

Reviewer #1 (Remarks to the Author):

The authors report on the effects of valproic acid-treated marmosets in the dmPFC with a focus on dynamics of spine formation and removal, generation of clustered spines, and possible effects of nasal application of oxytocin on these parameters. I found the manuscript highly interesting and relevant for ASD research given the model system used, the techniques applied and the questions addressed.

They report an increase in the synaptic dynamics of apical dendrites of L2/3 pyramidal neurons in the dmPFC, a brain area definitively associated with ASD. The turnover of dendritic spines was increased, in particular that of clustered spines. Overall this appears to be combined with an apparent increased survival of these spines which was dramatically higher compared to controls.

This increase in spine clustering is reduced after nasal application of oxytocin.

Technically the work is very challenging, and the longitudinal data of spine dynamics in the dmPFC using in vivo two-photon microscopy presented are of very high quality.

The analysis of VPA-exposed adult marmosets is rather diverse from the typical study of ASD in the developing brain. Studies in the adult might have some principle differences and its use in ASD research is to me unclear but some investigations show indeed that there is plasticity in the ASD phenotype in adults.

I am not sure of the relevance of a comparison of the spine generation rate and carry-over fraction of marmosets would have to be compared to a mouse model of ASD, here then MeCP2.

Oxytocin receptors and circuits are prominently expressed in the VTA and have been shown in mouse ASD models to be important here in this region, while the authors look here in the dmPFC, and show also expression of these receptors. However, to me the effects here in the dmPFC of marmosets appears minute, and most parameters of spine dynamics appear not affected by oxytocin. It is also not clear whether these applications have any effect on ASD-relevant behaviour, so I am not convinced entirely whether these data should be part of the paper.

The discussion is over-long and should be cut considerably.

Reviewer #2 (Remarks to the Author):

By using in vivo time-lapse two-photon microscopy, Noguchi and colleagues examined structural spine dynamics of tuft dendrites of upper-layer pyramidal neurons and adjacent axons in the dorsomedial prefrontal cortex in adult marmosets. This manuscript reports that valproic acid (VPA)-induced adult ASD marmoset model exhibits upregulated dendritic spine turnover, increased new spine formation in clusters, enhanced new spine survivorship, and upregulated local axonal bouton dynamics. Using intranasal administration of oxytocin, the authors also demonstrated that oxytocin reduced the tendency of spines to cluster without affecting general turnover rate of spines in VPA model. The manuscript is well written, the experiments are well-designed, and obtained data are largely convincing. However, there are several important loose ends the authors should address and additional control experiments are needed. To strengthen their findings, the authors should address the concerns listed below:

Major comments:

1. The authors should show whether there is any difference in spine dynamics including spine formation, elimination, and new spine survivorship between 3-day (day 0 to day 3, or day 3 to day 6) vs 6-day (day 0 to day 6) observation experiments. It is not clear why the authors report 6-day data in Fig 1 and 3-day results in other figures. Can authors conclude the same findings in Fig1 by using 3-day data only and/or in Fig 3-5 by 6-day data only?

2. The authors mention that both male and female animals were used. Please clarify if there are any gender differences in the current results. Specifically, oxytocin and oxytocin receptor expressions are known to be sex-specific.

3. In Figure 2, the authors claim that spine clustering reached a plateau at a distance of 3 μm between spines, but it is not clear if this distance effect is abolished with clustering threshold $> 3 \mu\text{m}$. Please show spine clustering data with different thresholds such as 6 μm and 9 μm .

4. In Figure 2, please show the comparison of the % of clustered spine elimination (same as Fig. 2B but for elimination). Does the distance only affect clustered spine generation not elimination? Any crosstalk between formation and elimination was observed? Heterosynaptic mechanisms should be discussed (PMIDs: 37829671, 25558061).

5. In Figure 3, please clarify the difference between carryover in panel C and spine stability of new spines in panel E. Are they conceptually the same observation? Please also show and compare the carryover and stability data from 3-day observation (short-term survivorship) vs 6-day observation (long-term survivorship).

6. Is axonal plasticity distance-dependent? In Figure 4, please show if presynaptic boutons also undergo dynamics in clusters similar to Figure 3.

7. What is the effect of nasal administration of oxytocin on spines in unexposed control marmosets? Can oxytocin alone induce spine dynamics, which might be abolished in VPA-exposed animals? Testing the physiological role of oxytocin in control marmosets is critical, as this is a big claim of the manuscript. In addition, there seems to be a trend towards a decrease in clustered spine generation and an increase in non-clustered spine generation. Are these experiments appropriately powered? What are the effects of oxytocin on survival fraction? Why are these clustered generation rates not being compared to UE and UE+OT?

Minor comments:

1. Please clarify their current VPA ASD marmoset model is induced by prenatal VPA exposure. This should be clearly shown in the experimental schema and timeline in Fig1A. In addition, the authors should discuss if their findings are specific to their method, in that only prenatal VPA, but not postnatal VPA

exposure, leads to the observed spine and axon dynamics.

2. In figure 4, the authors show a change in the bouton density proportion between ipsi- and contra-lateral following VPA-induced ASD model. However, it is unclear from the text and graph, whether this is accompanied by a statistically significant increase of contralateral bouton density and a decrease in ipsilateral bouton density between groups. This may have interesting functional implications for microcircuit activity.

3. In figure 3, the authors provide evidence of an increased surviving fraction of newly generated spines. As a reader, one might expect that this increase in surviving fraction would result in an increase in spine density, yet this is not the case. This suggests the increase elimination compensates for the increase in surviving fraction. This is a very interesting finding; however, this raises a fundamental question on the biological impact of this discovery. What is the functional effect of this increase in turnover if it is not leading to an overall increase in spine density? Overall weaker (more new) synapses? Fewer/more functional synapses despite spine density?

Point-by-point responses to the reviewers:

We express our sincere gratitude to the reviewers for dedicating their valuable time to critically assess and comprehend the content of our paper. Their thoughtful review and understanding of our results report are greatly appreciated.

Reviewer #1 (Remarks to the Author):

The authors report on the effects of valproic acid-treated marmosets in the dmPFC with a focus on dynamics of spine formation and removal, generation of clustered spines, and possible effects of nasal application of oxytocin on these parameters. I found the manuscript highly interesting and relevant for ASD research given the model system used, the techniques applied and the questions addressed.

We thank Reviewer#1 for considering our results as both highly interesting and relevant for ASD research.

They report an increase in the synaptic dynamics of apical dendrites of L2/3 pyramidal neurons in the dmPFC, a brain area definitively associated with ASD. The turnover of dendritic spines was increased, in particular that of clustered spines. Overall this appears to be combined with an apparent increased survival of these spines which was dramatically higher compared to controls.

This increase in spine clustering is reduced after nasal application of oxytocin.

Technically the work is very challenging, and the longitudinal data of spine dynamics in the dmPFC using in vivo two-photon microscopy presented are of very high quality.

The analysis of VPA-exposed adult marmosets is rather diverse from the typical study of ASD in the developing brain. Studies in the adult might have some principle differences and its use in ASD research is to me unclear but some investigations show indeed that there is plasticity in the ASD phenotype in adults.

1. The population of autistic adults is growing rapidly. They have multiple problems in their daily lives including anxiety, difficulties with verbal and nonverbal social communications, and sensory problems despite having undergone behavioral therapy and other interventions during their developmental years. We believe it is desirable to have more treatment options for people after puberty in addition to those addressed in childhood. We add following sentences in the Introduction section:

(L.76) A notable portion of individuals with ASD persists in experiencing a range of challenges such as anxiety or depression in their daily lives into adulthood. It is critical to understand ASD

pathophysiology in adult ASD model animals and explore treatments.

I am not sure of the relevance of a comparison of the spine generation rate and carry-over fraction of marmosets would have to be compared to a mouse model of ASD, here then MeCP2.

2. We have considered the possibility that readers should want to know if there is a characteristic link between our marmoset model and an existing animal model (in this case, the mouse). However, the comparison with the specific mouse model may confuse the potential readers as Reviewer #1 suggested. Therefore, we would like to remove previous Figures S2B and S2C and the following sentences in the main text:

(L. 199 of previous manuscript version) Lastly, we compared the clustering of VPA-exposed marmosets with those of a mouse model. ...Thus, the rate of spine clustering in the marmoset model differed markedly in comparison to the MeCP2 duplication mouse model.

Oxytocin receptors and circuits are prominently expressed in the VTA and have been shown in mouse ASD models to be important here in this region, while the authors look here in the dmPFC, and show also expression of these receptors. However, to me the effects here in the dmPFC of marmosets appears minute, and most parameters of spine dynamics appear not affected by oxytocin. It is also not clear whether these applications have any effect on ASD-relevant behaviour, so I am not convinced entirely whether these data should be part of the paper.

The discussion is over-long and should be cut considerably.

3. As noted by Reviewer #1, it is known that oxytocin receptors are strongly expressed including VTA, basal ganglia, nucleus accumbens, and hypothalamus, while their expression in the cortex is relatively low. Therefore, it is possible that indirect effects from those nuclei may be exerted on the cortex. On the other hand, oxytocin receptors in the prefrontal cortex are detectable and may play a role (Fig. S6). As shown in the new Fig. 5, oxytocin also altered the bias of clustered dendritic spine generation in UE marmosets. (but in the opposite direction from VPA-exposed animals). This finding underscores the conclusion that oxytocin administration induces changes in synaptic properties within the prefrontal cortex.

In response to the suggestion of Reviewer #1, we have revised the Discussion section for conciseness. Specifically, the following statements were deleted (displayed by the line number of the previous manuscript):

(L. 328) Clustered emergent spines that receive structurally or functionally similar inputs

are thought to enable nonlinear dendritic computation and may facilitate the ability to acquire memory and behavioral skills through learning and experience ^{11, 12, 14-17, 67}. Alternatively, clustered generated spines may also help stabilize formed circuits and accelerate learning due to their redundant involvement in the circuit ⁶⁸. In fact,

(L. 347) Compared to VPA-exposed marmosets, the degree of clustered emergence in *MeCP2* duplication mice appears to be substantially lower. The reasons for this discrepancy are currently unclear. However, it may be due to differences in spine density, plasticity molecules, or subcellular organelles between species, cortical areas, syndromic and idiopathic ASD models, and deep and upper-layer pyramidal neurons. In any case, the strong clustering tendency in the dmPFC of VPA-exposed marmosets underscores the importance of clustering in the pathogenesis of idiopathic ASD and warrants further research of clustered spine emergence in both physiological and pathophysiological conditions.

(L. 375) It will be interesting to analyze whether mismatches in plasticity control involving local and long-range connections facilitate autonomous local circuit remodeling independent of the highly contextualized information computed in the brain-wide global network.

(L. 389) Oxytocin receptors in marmosets are notably expressed in the basal ganglia of the nucleus basalis of Meynert (NBM), a prominent cholinergic region within the brain. The NBM serves as a significant source of cholinergic innervation to various brain regions, including the neocortex, and plays a pivotal role in regulating selective attention and motivational processes⁵⁴. It is plausible that the alteration of spine clustering induced by oxytocin may involve the modulation of cholinergic inputs. Oxytocin's inhibitory effect on spine clustering could restrain excessively prolonged circuit stability and mitigate behavioral perseverance, as reported in human ASD ⁷⁸.

(L. 406) Suppression of excessive microglial activation with minocycline or other matrix metalloproteinase-9 inhibitors has been reported to suppress excessive spine turnover in fragile X model mice ⁸. Our transcriptomic analysis also revealed significant downregulation of myelin-related genes, such as PLP1, which suppresses heightened cortical plasticity (see Supplementary Data 1). On the other hand, it has been shown that the stress hormone corticosterone increases spine turnover in mice ⁸¹.

Reviewer #2 (Remarks to the Author):

By using in vivo time-lapse two-photon microscopy, Noguchi and colleagues examined structural spine dynamics of tuft dendrites of upper-layer pyramidal neurons and adjacent axons in the dorsomedial prefrontal cortex in adult marmosets. This manuscript reports that valproic acid (VPA)-induced adult ASD marmoset model exhibits upregulated dendritic spine turnover, increased new spine formation in clusters, enhanced new spine survivorship, and upregulated local axonal bouton dynamics. Using intranasal administration of oxytocin, the authors also demonstrated that oxytocin reduced the tendency of spines to cluster without affecting general turnover rate of spines in VPA model. The manuscript is well written, the experiments are well-designed, and obtained data are largely convincing. However, there are several important loose ends the authors should address and additional control experiments are needed. To strengthen their findings, the authors should address the concerns listed below:

We thank Reviewer #2 for dedicating time to thoroughly review our manuscript. We are deeply honored to have received not only a comprehensive understanding of the manuscript but also a positive evaluation of its experimental design and the results derived from it.

Major comments:

1. The authors should show whether there is any difference in spine dynamics including spine formation, elimination, and new spine survivorship between 3-day (day 0 to day 3, or day 3 to day 6) vs 6-day (day 0 to day 6) observation experiments. It is not clear why the authors report 6-day data in Fig 1 and 3-day results in other figures. Can authors conclude the same findings in Fig1 by using 3-day data only and/or in Fig 3-5 by 6-day data only?

1. For the previous Figures 1F and 1G, a 6-day observation period was initially employed, while a 3-day period was utilized for the remaining figures. Nevertheless, recognizing the potential confounding effects of mixed observation periods on figure interpretation, we harmonized the observation period to 3 days. Consequently, our analysis yielded essentially identical results to those obtained during the 6-day period (Revised Figs. 1F and 1G).

2. The authors mention that both male and female animals were used. Please clarify if there are any gender differences in the current results. Specifically, oxytocin and oxytocin receptor expressions are

known to be sex-specific.

2. In reference to Figure 5, we analyzed whether there were sex differences in dendritic spine generation or elimination following oxytocin administration to VPA-exposed animals (Supplementary Figures S9-S12). The scatterplot data for each dendrite shown separately for UE and VPA-exposed males and females. As a result, we could not detect statistically significant sex differences.

We modified the manuscript as follows:

(L. 303) It is known that there is sexual dimorphism in oxytocin receptor expression that well investigated in rodents. On the other hand, it has been reported that an absence of sex differences in OT-immunoreactive neurons in marmoset brain regions such as paraventricularis and supraopticus of the hypothalamus as well as in the bed nucleus of the stria terminalis and the medial amygdala. We therefore examined the effect of sexual differences on the effects of oxytocin by displaying scatter plots of the generation or elimination rate of spines before and after oxytocin administration (Supplementary Figure S9-S12). The scatterplot data for each dendrite shown separately for UE and VPA-exposed males and females. We could not detect statistically significant sex differences in VPA-exposed marmosets with respect to total, clustered, and non-clustered spine generation/elimination, potentially due to the considerable variation in values. Similarly, in the UE marmosets, although we detected statistically significant sex differences in several items (total and clustered spine generation at D6-9, and total spine elimination at D0-3), we could not find any systematically comprehensible sex differences across the parameters under investigation.

Furthermore, in the Discussion Limitations section, we stated the following:

(L. 455) We could not conclude a systematic sex difference in the present data (depicted in Figs. S11-S14 for oxytocin treatment).

3. In Figure 2, the authors claim that spine clustering reached a plateau at a distance of 3 μm between spines, but it is not clear if this distance effect is abolished with clustering threshold $> 3 \mu\text{m}$. Please show spine clustering data with different thresholds such as 6 μm and 9 μm .

3. We revised Supplementary Fig. S2 and main text as follows:

(L. 164) We explored the impact of adjusting cluster thresholds to 3, 6, and 9 microns on the outcomes of spine generation and elimination clustering (Figure S2C–P). The clustering probability of the generated spines actually observed in VPA-exposed animals corresponded to a probability of $P = 0.00042$ for the distribution of clustering probabilities predicted by the simulation at the 3 μm threshold (Figure S2D). This P-value is much smaller than the

probabilities at the 6 μm ($P = 0.015$) or 9 μm ($P = 0.25$) thresholds (Figures S2F and S2H). To investigate this further, we plotted the actual clustering probabilities against the simulated clustering probability, and again found that the difference between chance-level clustering and observed clustering was greatest at the 3 μm threshold that provides the Youden's index in the chart (Fig. S2O)³⁹. Since the specific clustering bias disappears at the 9 μm threshold, indicating a physiological significance of the interaction within this distance. This analysis elucidates that the threshold set at 3 μm offers a good cut-off value for assessing spine clustered generation. In contrast, the clustering probability of the generated spine pairs was remained below the 95th percentile in the UE group (Figure 2E).

4. In Figure 2, please show the comparison of the % of clustered spine elimination (same as Fig. 2B but for elimination). Does the distance only affect clustered spine generation not elimination? Any crosstalk between formation and elimination was observed? Heterosynaptic mechanisms should be discussed (PMIDs: 37829671, 25558061).

4. We conducted an analysis of clustered spine elimination and prepared a revised Fig. S3 that incorporates the previous Fig. 2G and H. Although there is a disparity in the effect of distance on clustered spine elimination between UE and VPA-exposed animals (revised Fig. S3C), this effect was considerably less pronounced than spine generation (Fig. 2C):

(L. 177) On the other hand, the specific clustering of eliminated spines was not as prominent as generated spines either in UE and VPA-exposed animals (Figures S2I-N, S2P, and S3). Although there is a disparity in the effect of distance on clustered spine elimination between UE and VPA-exposed animals (Figure S3C), this effect was considerably less pronounced than spine generation (Figure S2B). The slight increase in clustered elimination observed in VPA-exposed marmosets could stem from heightened nonspecific clustering associated with the larger number of eliminated spines. Simulation analysis revealed that the clustering bias of spine elimination in VPA-exposed animals is not significant (Figures S2P, S3D, and S3E).

We modified the main text about the crosstalk between spine formation and elimination as follows:

(L. 186) Furthermore, an analysis of the crosstalk between spine generation and elimination was conducted, as depicted in Figure S4. Cumulative frequency distributions were generated for the distances between newly generated spines and their nearest eliminating spines. Consequently, no statistically significant difference was observed between the UE and VPA-exposed marmosets in this regard (Figure S4).

Regarding the heterosynaptic mechanism identified in the present experimental setup, we have incorporated the following sentences into the Discussion section:

(L. 376) There are several studies that have shown that synaptic plasticity diffuses from stimulated spines into neighboring spines and interacts there^{5, 34, 39, 63-66}. The relationship between heterosynaptic plasticity and the subsequent new generation or elimination of surrounding spines is a subject for future work.

5. In Figure 3, please clarify the difference between carryover in panel C and spine stability of new spines in panel E. Are they conceptually the same observation? Please also show and compare the carryover and stability data from 3-day observation (short-term survivorship) vs 6-day observation (long-term survivorship).

5. We revised Figure 3 and main text to clarify the difference between carryover and spine stability (spine “stability” will be referred to as “survival rate” hereafter) of new spines:

(L. 200) The carryover spines were defined as the newly generated spines that survived to the last session. In other words, carryover spines are spines whose presence was denied by the first observation and confirmed by the second and third observations (Figures 3A and 3B, GS). As discussed below, the survival rate of carryover spines is not different between UE and VPA-exposed animals. However, since the fraction of newly generated spines are two times higher in VPA-exposed animals (Figure 1D), the fraction of carryover spines was two times higher in VPA-exposed marmosets than in UE marmosets (Figure 3C).

(L. 214) By calculating the carryover fraction, we can understand how newly generated spines contribute to the overall synaptic population. Next, by calculating the spine survival rate, we can focus on the persistence of each newly generated and pre-existing spine, offering a more targeted measure of synaptic resilience. Spine survival rate was defined as the percentage of spines present at the second observation that were still present at the third observation (Figure 3B).

“Carryover spines” are spines whose presence was denied by the first two-photon microscopy observation and confirmed by the second and third observations. Therefore, we cannot show long-term (6-day) survivorship of the carryover spines.

6. Is axonal plasticity distance-dependent? In Figure 4, please show if presynaptic boutons also undergo dynamics in clusters similar to Figure 3.

6. Due to the considerably lower density of boutons compared to dendritic spines in the sample utilized for this investigation, we observed a relatively limited occurrence of simultaneous bouton generation or elimination within the measured axons. Consequently, it was not possible to create a cumulative distribution with a high degree of confidence (N = 6, 6, 5, and 10 axons featuring simultaneously

generated bouton pairs, and N = 5, 1, 4, and 5 axons featuring simultaneously eliminated bouton pairs, out of a total of 28, 21, 22, and 25 UE-ipsi, UE-contra, VPA-ipsi, and VPA-contra axons, respectively). Consequently, we refrain from incorporating this data into the main text. We added the following to the Results section:

(L. 307) Due to the lack of enough coincident events, we were unable to analyze the interactions between newly generated or eliminated axonal boutons.

7. What is the effect of nasal administration of oxytocin on spines in unexposed control marmosets? Can oxytocin alone induce spine dynamics, which might be abolished in VPA-exposed animals? Testing the physiological role of oxytocin in control marmosets is critical, as this is a big claim of the manuscript. In addition, there seems to be a trend towards a decrease in clustered spine generation and an increase in non-clustered spine generation. Are these experiments appropriately powered? What are the effects of oxytocin on survival fraction? Why are these clustered generation rates not being compared to UE and UE+OT?

7. We express our gratitude for the opportunity to investigate the effects of oxytocin administration on the clustered generation or elimination of dendritic spines in UE marmosets. As demonstrated in the revised Fig. 5, our analyses indicate that oxytocin administration induces alterations in clustered spine generation even in UE marmosets. Notably, oxytocin tended to promote clustered spine generation, contrary to the case in VPA-exposed animals. The underlying mechanism remains elusive and warrants further investigation, suggesting that synaptic function is quite different in UE and VPA-exposed animals.

Regarding the statistical power to detect clustered spine generation or elimination, it is pertinent to note the substantial variability present within each dendrite, encompassing instances where no clusters are observed. Consequently, we were unable to discern statistically significant changes in the average occurrence of clustered spine generation (Fig. 5C) in either UE or VPA-exposed animals. To address this limitation, we employed simulation analysis utilizing a model that factors in dendrite length and the number of spines, offering enhanced statistical power (refer to Figure 5E-K and Figure S7D-J). In relation to the above, we modified the Results section (L. 290~302) accordingly.

Minor comments:

1. Please clarify their current VPA ASD marmoset model is induced by prenatal VPA exposure. This should be clearly shown in the experimental schema and timeline in Fig1A. In addition, the authors should discuss if their findings are specific to their method, in that only prenatal VPA, but not postnatal VPA exposure, leads to the observed spine and axon dynamics.

1. In accordance with the suggestion provided by Reviewer #2, we have made revisions to Fig. 1A to enhance the clarity of the timeline depicting VPA exposure and the two-photon imaging.

We add the following sentences in the Results section:

(L. 104) We employ the prenatal valproic acid exposure model due to its relevance to in utero VPA exposure in humans and a rich accumulation of findings across multiple animal species.

2. In figure 4, the authors show a change in the bouton density proportion between ipsi- and contra-lateral following VPA-induced ASD model. However, it is unclear from the text and graph, whether this is accompanied by a statistically significant increase of contralateral bouton density and a decrease in ipsilateral bouton density between groups. This may have interesting functional implications for microcircuit activity.

2. The statistical analysis method in Fig. 4D was changed to two-way ANOVA followed by Fisher's LSD post hoc test to examine whether the increase in contralateral bouton density and the decrease in ipsilateral bouton density in VPA-exposed animals were statistically significant or not. The results showed that they were not statistically significant at a significance level of 0.05 (New Figure 4D). The smaller difference in bouton densities between ipsilateral and contralateral in VPA-exposed animals than in UE-animals is the sum of these statistically insignificant effects.

3. In figure 3, the authors provide evidence of an increased surviving fraction of newly generated spines. As a reader, one might expect that this increase in surviving fraction would result in an increase in spine density, yet this is not the case. This suggests the increase elimination compensates for the increase in surviving fraction. This is a very interesting finding; however, this raises a fundamental question on the biological impact of this discovery. What is the functional effect of this increase in turnover if it is not leading to an overall increase in spine density? Overall weaker (more new) synapses? Fewer/more functional synapses despite spine density?

3. Thank you for raising this important point. It is well-established in the literature that spine turnover tends to be more pronounced in younger animals (Holtmaat et al., 2006, <https://doi.org/10.1038/nature04783>; Zuo et al., 2006, <https://doi.org/10.1016/j.neuron.2005.04.001>). Additionally, studies have reported elevated spine turnover in a mouse model characterized by high learning ability (Frank et al., 2018, <https://doi.org/10.1038/s41467-017-02751-2>). Mouse experiments have further demonstrated increased spine turnover following learning tasks (Ma et al., 2022, <https://doi.org/10.1016/j.semdb.2021.05.015>; Chen et al., 2015, <https://doi.org/10.1038/nn.4049>; Ma et al., 2016, <https://doi.org/10.1002/dneu.22313>). The generation of new dendritic spines shows the neogenesis of new synaptic connections (i.e., new neuronal circuits). It may be possible to eliminate some spines, keeping the overall spine density constant and preventing excessive neuronal activity,

while approaching a more optimized neural circuit for the environment.

In light of our results, we add these sentences in the Discussion section:

(Line 380) The observed higher spine turnover, or more precisely, the larger population of spines with heightened turnover, may suggest the potential for the formation of more efficient neuronal circuits (Runge et al. 2020, <https://doi.org/10.3389/fnsyn.2020.00036>).

Updated figures

Figure 1. In vivo two-photon imaging of mature marmoset dendrites. (A) A marmoset was anesthetized for two-photon (2P) imaging. The lower panel shows the experimental schema. (B) Longitudinal 2P imaging from the PFC layer-2/3 pyramidal neuron tuft dendrites. Every 3 days, the same dendrites were imaged in UE (unexposed; control animals) and VPA-exposed (ASD model) marmosets, and the same spines were labeled using the same numbers throughout the period. (C) Mean spine densities of dendrites (mean \pm s.e.m.; $P = 0.85$, Mann-Whitney U test; $n = 14$ and 12 dendrites in four and three UE and VPA-exposed animals, respectively). (D) Three-day spine generation rates (mean \pm s.e.m.; $P = 0.048$, Mann-Whitney U test; the same dendrites as in (C)), and 3-day elimination ($P = 0.037$). (E) Cumulative plots of the normalized spine volume distribution ($P = 0.49$; Kolmogorov-Smirnov test; $n = 480$ and 473 spines in UE and VPA-exposed animals, respectively). (F) Cumulative plots of eliminated and surviving spines in UE animals ($P = 0.0035$; $n = 36$ and 444 spines, respectively), and in VPA-exposed animals during the 3-day observation ($P < 0.001$; $n = 56$ and 417 spines, respectively). (G) The population of eliminated and surviving spines during the 3-day observation are shown separately for groups with a spine volume of $V < 0.15$ and $V > 0.15$ ($V < 0.15$; $P = 0.0019$; Fisher's exact test; $n = 25, 46, 210$, and 169 spines for eliminated UE, eliminated VPA-exposed, surviving UE, and surviving VPA-exposed spines, respectively) ($V > 0.15$; $P = 0.83$; $n = 11, 10, 234$, and 248 spines, respectively). *** $P < 0.001$; ** $P < 0.01$; * $P < 0.05$; n.s., not significant.

Points of modifications:

Fig. 1A: VPA exposure was added to the experimental time course.

Fig. 1C, D: Individual data points were added to the bar charts.

Fig. 1F, G: Observation period was changed to 3-day.

Figure 2. Dendritic spine clustered generation is more predominant in VPA-exposed marmosets. (A) Schematic drawing of clustered spine generation. Pairs of newly generated or eliminated spines were considered to be clustered if they occurred within 3 μm of each other. Spines colored magenta or gray represent clustered and non-clustered spines, respectively. (B) Comparison of the percentage of clustered generated spines to the total number of spines between UE and VPA-exposed animals (mean \pm s.e.m.; $P = 0.026$, Mann-Whitney U test; $n = 14$ and 12 dendrites in four UE and three VPA-exposed animals, respectively). (C) Distribution of inter-spine distances between newly generated spines ($P = 0.0009$; Kolmogorov-Smirnov test; $n = 45$ and 85 spine pairs in UE and VPA-exposed animals, respectively). Magenta-shaded area indicates inter-spine distances shorter than 3 μm , and numbers indicate the probabilities of clustering (within 3 μm). To prevent underestimation of the inter-spine distance, dendrites were concatenated into one long dendrite. (D) Validation of clustering bias by Monte Carlo simulation. In the simulation, the new spine positions were randomly determined with a uniform distribution over the summed dendrite length. Clustering probabilities for all inter-spine distances were calculated for each simulation, and the distributions of the clustering probability from 100,000 iterations are shown in (E and F). (E, F) Circles connected with gray lines represent probability plots of clustering events from 100,000 simulations; the actual numbers of spine clusters are represented by arrows ($P = 0.23$ and 0.00042; $n = 44$ and 86 newly generated spines in 14 and 12 dendrites in UE and VPA-exposed animals, respectively). Dotted red lines show 95th percentiles. *** $P < 0.001$; * $P < 0.05$; n.s., not significant.

Points of modifications:

Fig. 2B: Individual data points were added to the bar charts.

Fig. 2G, H: Moved to Figure S3D, E

Figure 3. The surviving fraction of newly generated spines (carryover spines) is much larger in VPA-exposed marmosets than in controls, although spine stability is similar. (A) Representative dendrite images taken every 3 days are presented as in Fig. 1B. (B) A diagram showing four patterns of spine generation and elimination. The red dot in the legend indicates the carryover spines labeled with a red dot in Figs. 1B and 3A. (C) The ratio of the surviving fraction of newly formed spines (carryover spines) to the total spines was larger in VPA-exposed animals than in UE animals. ($P = 0.037$, Fisher's exact test; $n = 13, 448, 22$, and 355 for UE carryover spines, UE total spines, VPA-exposed carryover spines, and VPA-exposed total spines, respectively). (D) Among clustered generated spines, the ratio of the carryover spine fraction to the total spines was much larger in VPA-exposed animals than in UE animals ($P = 0.0022$, Fisher's exact test; $n = 3, 448, 14$, and 355 for UE carryover, UE total, VPA-exposed carryover, and VPA-exposed total spines, respectively). By contrast, the carryover spine fraction was not significantly different among non-clustered spines ($P > 0.99$, Fisher's exact test; $n = 10, 448, 8$, and 355 for UE carryover, UE total, VPA-exposed carryover, and VPA-exposed total spines, respectively). (E) The survival rate of each newly generated spine at Day-3 was not significantly different between UE and VPA-exposed animals (mean \pm s.e.m.; $P = 0.94$; Mann-Whitney U test; $n = 13$ and 10 dendrites from three UE and three VPA-exposed animals, respectively). (F) The survival rate of each pre-existing spine at Day-3 was not significantly different between UE and VPA-exposed animals (mean \pm s.e.m.; $P = 0.091$). (G) The survival rates of newly generated spine at Day-3 were analyzed separately for the clustered and non-clustered spines ($P > 0.99$, Fisher's exact test; $n = 3, 10, 3$, and 7 for clustered GS, non-clustered GS, clustered GL, and non-clustered GL spines, respectively, in UE animals) ($P > 0.99$, Fisher's exact test; $n = 14, 8, 14$, and 6 , respectively, in VPA-exposed animals). ** $P < 0.01$; * $P < 0.05$; n.s., not significant.

Points of modifications:

Fig. 3B: The symbols were changed to a spine-shape.

Fig. 3C-G: Graph titles were added. Graph y-axis labels were changed.

Fig. 3E, F: Individual data points were added to the bar charts.

Figure 4. Two-photon in vivo axon imaging in ASD model marmosets. (A) AAV vectors expressing different colored fluorescent proteins were inoculated into each hemisphere of the dmPFC. (B) Axons from the contralateral hemisphere had green fluorescence, and axons from the ipsilateral hemisphere and dendrites had red fluorescence. (C) A representative axon from a VPA-exposed animal (the axon surrounded by the yellow rectangle in (B)) was imaged every 3 days. (D) Mean bouton densities are significantly larger in ipsilateral than in contralateral axons in UE animals (UE-ipsi vs. UE-contra, $P = 0.015$; $n = 28$ and 21 ipsilateral and contralateral axons, respectively, from three animals; two-way ANOVA followed by Fisher's LSD *post hoc* test). They are not significantly different between ipsilateral and contralateral axons in VPA-exposed animals (VPA-ipsi vs. VPA-contra, $P = 0.39$; $n = 22$ and 25 ipsilateral and contralateral axons, respectively, from two animals). The difference in bouton density between the ipsilateral axons or between the contralateral axons was not statistically significant (UE-ipsi vs. VPA-ipsi, $P = 0.56$; UE-contra vs. VPA-contra, $P = 0.32$). (E) The bouton size distribution between these animals ($P = 0.57$; Kolmogorov-Smirnov test; $n = 149$ and 107 boutons in 49 and 47 axons in three and two UE and VPA-exposed animals, respectively). (F) There is a significant difference in three-day bouton generation rates between these animals ($P = 0.023$, Mann-Whitney U test; $n = 49$ and 47 axons in three and two UE and VPA-exposed animals, respectively), and also in 3-day bouton elimination ($P = 0.033$). (G, H) Three-day mean bouton generation rates (G) (mean \pm s.e.m.; $P = 0.018$ and $P = 0.49$, Mann-Whitney U test; $n = 28$, 22 , 21 , and 25 for UE ipsilateral, VPA-exposed ipsilateral, UE contralateral, and VPA-exposed contralateral axons, respectively) and elimination rates (H) ($P = 0.094$ and $P = 0.28$) for each axon type. * $P < 0.05$; n.s., not significant.

Points of modifications:

Fig. 4D: Statistical analysis method was changed to 2-Way ANOVA.

Fig. 4D, E, G, H: Individual data points were added to the bar charts.

Figure 5. Oxytocin modifies spine generation proximity in UE and VPA-exposed marmosets.

(A) Schematics of oxytocin nasal administration and two-photon (2P) imaging (Left panel) and transnasal administration using a micropipette (Right panel). Saline was given to marmosets in the green-shaded period, while oxytocin was given in the magenta-shaded period. (B) Three-day total spine generation rates (mean \pm s.e.m.; Two-way ANOVA, group: $P = 0.011$, time: $P = 0.42$, interaction: $P = 0.80$; *post hoc* Tukey's test, D0-3 (UE vs. VPA): $P = 0.035$; $n = 15$ dendrites from four UE animals; $n = 12$ dendrites from three VPA-exposed animals). (C) Three-day clustered spine generation rates (mean \pm s.e.m.; Two-way ANOVA, group: $P = 0.085$, time: $P = 0.63$, interaction: $P = 0.30$; *post hoc* Tukey's test, D0-3 (UE vs. VPA): $P = 0.038$; $n = 15$ dendrites from four UE animals; $n = 12$ dendrites from three VPA-exposed animals). (D) Three-day non-clustered spine generation rates (mean \pm s.e.m.; Two-way ANOVA, group: $P = 0.015$, time: $P = 0.51$, interaction: $P = 0.30$; *post hoc* Tukey's test, D6-9 (UE vs. VPA): $P = 0.017$; $n = 15$ dendrites from four UE animals; $n = 12$ dendrites from three VPA-exposed animals). (E–J) Simulation analysis to analyze the effects of oxytocin on clustering bias of newly generated spines in the UE (E–G) and VPA-exposed (H–J) animals ($n = 15$, 24, and 15 newly generated spine pairs from four UE animals, and $n = 32$, 32, and 28 pairs from three VPA-exposed animals, during the D0–3, D3–6, and D6–9 periods, respectively). Dotted red lines show 95th percentiles. (K) The P-values expressed in logarithm from (E)–(J) are indicated. ** $P < 0.01$; * $P < 0.05$; n.s., not significant.

Points of modifications:

Analysis results in UE animals were added.

Fig. 5B-D: Statistical analysis method was changed to 2-Way ANOVA.

Fig. 5B-D: Individual data points were added to the bar charts.

REVIEWERS' COMMENTS:

Reviewer #1 (Remarks to the Author):

My comments have been properly addressed, and I do support of this nice piece of work in CommBiol.

Reviewer #2 (Remarks to the Author):

The authors fully addressed my review comments.

I only have one suggestion regarding the title of Fig. 4. A more conclusive title rather than methodological one would be better for the readers, as a higher turnover rate of axons is a very interesting finding.

There was an error on the y-axis label on Fig. 3G.

Point-by-point responses to the reviewers:

We extend our sincere gratitude to the reviewers for their feedback on the manuscript. We have fixed all the points raised by the reviewers. Furthermore, the figures and text of the manuscript have been revised to address editorial requests (such as visibility for color blindness, etc.).

Reviewer #1 (Remarks to the Author):

My comments have been properly addressed, and I do support of this nice piece of work in CommBiol. Thank you very much for your comment on the manuscript. We appreciate the valuable time you spent on the peer review process.

Reviewer #2 (Remarks to the Author):

The authors fully addressed my review comments.

I only have one suggestion regarding the title of Fig. 4. A more conclusive title rather than methodological one would be better for the readers, as a higher turnover rate of axons is a very interesting finding. There was an error on the y-axis label on Fig. 3G.

Thank you very much for your comments on the manuscript. We appreciate the valuable time you spent on the peer review process.

We revised the title of Fig. 4 as follows: A projection-specific elevation in synaptic turnover rate in VPA-exposed marmoset axons.

We have fixed the error on the y-axis label on Fig. 3G.

—

Other revisions:

Revisions of main text:

L. 10-13: The affiliated organizations' addresses were changed from the prefecture name to the city name.

L. 20: The Abstract was revised to the present tense.

L. 63, L433, L451: The use of the word "significant", which does not refer to statistical test results, has been replaced.

L. 258: "...contralateral bouton gain (Fig. 4g)." was revised to "...contralateral bouton gain (Fig. 4g) or loss (Fig. 4h)."

L. 478, 480, and 481: In the Author Contributions' section, the three authors' initials were modified to distinguish them.

L. 1101, 1152: Figure captions/legends were revised in Figure 4 and Figure 6.

Figures and Supplementary figures:

Figure numbering labels have been modified from upper case letters to lower case letters.

(Figures and main text)

A change was made to the color and shape of the symbols to ensure that the figures are clearly identifiable when printed in black and white, as well as being understandable by users with color blindness.

To number the supplementary figures in the order of appearance in the main text, Supplementary Figures 8, 9, 10, 11, and 12 are renumbered as Supplementary Fig. 12, 8, 9, 10, and 11, respectively.

Source data for Supplementary Figure 6 are provided.

Graph titles: Supplementary Figure 1, 2, 5, and 6.